# Refining Diffusion Planner for Reliable Behavior Synthesis by Automatic Detection of Infeasible Plans

**Kyowoon Lee**[*]
UNIST
leekwoon@unist.ac.kr

**Seongun Kim**[*]
KAIST
seongun@kaist.ac.kr

**Jaesik Choi**
KAIST, INEEJI
jaesik.choi@kaist.ac.kr

## Abstract

Diffusion-based planning has shown promising results in long-horizon, sparse-reward tasks by training trajectory diffusion models and conditioning the sampled trajectories using auxiliary guidance functions. However, due to their nature as generative models, diffusion models are not guaranteed to generate feasible plans, resulting in failed execution and precluding planners from being useful in safety-critical applications. In this work, we propose a novel approach to refine unreliable plans generated by diffusion models by providing refining guidance to error-prone plans. To this end, we suggest a new metric named *restoration gap* for evaluating the quality of individual plans generated by the diffusion model. A restoration gap is estimated by a *gap predictor* which produces *restoration gap guidance* to refine a diffusion planner. We additionally present an attribution map regularizer to prevent adversarial refining guidance that could be generated from the sub-optimal gap predictor, which enables further refinement of infeasible plans. We demonstrate the effectiveness of our approach on three different benchmarks in offline control settings that require long-horizon planning. We also illustrate that our approach presents explainability by presenting the attribution maps of the gap predictor and highlighting error-prone transitions, allowing for a deeper understanding of the generated plans.

## 1 Introduction

Planning plays a crucial and efficient role in tackling decision-making problems when the dynamics are known, including board games and simulated robot control (Tassa et al., 2012; Silver et al., 2016, 2017; Lee et al., 2018). To plan for more general tasks with unknown dynamics, the agent needs to learn the dynamics model from experience. This approach is appealing since the dynamics model is independent of rewards, enabling it to adapt to new tasks in the same environment, while also taking advantage of the latest advancements from deep supervised learning to employ high-capacity models.

The most widely used techniques for learning dynamics models include autoregressive forward models (Deisenroth & Rasmussen, 2011; Hafner et al., 2019; Kaiser et al., 2020), which make predictions based on future time progression. Although an ideal forward model would provide significant benefits, there is a key challenge that the accuracy of the model directly affects the quality of the plan. As model inaccuracies accumulate over time (Ross & Bagnell, 2012; Talvitie, 2014; Luo et al., 2019; Janner et al., 2019; Voelcker et al., 2022), long-term planning using imprecise models might yield sub-optimal performances compared to those achievable through model-free techniques. Building upon the latest progress in generative models, recent studies have shown promise in transforming reinforcement learning (RL) problems into conditional sequence modeling, through the modeling of the joint distribution of sequences involving states, actions, and rewards

---

[*]Equal Contribution
Codes are available at https://github.com/leekwoon/rgg.

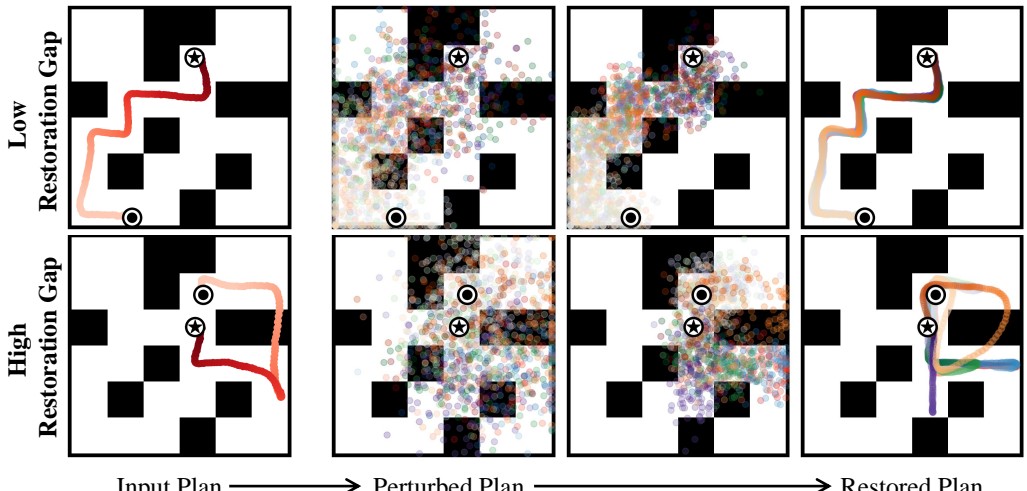

Figure 1: Illustration of two plans with low/high restoration gaps with a specified start ◉ and goal ★. For each input plan, we first perturb it using Gaussian noise. We then remove the noise from the perturbed plan by simulating the reverse SDE which progressively transforms the perturbed plan into the initial plan by utilizing the score function (Section 2.2). The restoration gap is then computed as the expected $L_2$ distance between the input plan and the plan restored from noise corruption (Section 3). The top example exhibits a smaller restoration gap because of its successful restoration close to the original plan, while the bottom example has a larger restoration gap due to its poor restoration performance. Plans restored from various noise corruptions are differentiated by distinct colors.

(Lambert et al., 2021; Chen et al., 2021; Janner et al., 2021, 2022). For instance, Diffuser (Janner et al., 2022) introduces an effective framework for generating trajectories using a diffusion model with flexible constraints on the resulting trajectories through reward guidance in the sampling phase. Although these approaches have achieved notable performance on long-horizon tasks, they still face challenges in generating outputs with unreliable trajectories, referred to as artifacts, resulting in limited performance and unsuitability for deployment in safety-critical applications.

This paper presents an orthogonal approach aimed at enhancing the plan quality of the diffusion model. We first propose a novel metric called *restoration gap* that can automatically detect whether generated plans are feasible or not. We theoretically analyze that it could detect artifacts with bounded error probabilities under regularity conditions. The restoration gap directly evaluates the quality of generated plans by measuring their restorability through diffusion models in which plans are exposed to a certain degree of noise, as illustrated in Figure 1. A restoration gap is estimated by a function approximator which we name a *gap predictor*. The gap predictor provides an additional level of flexibility to the diffusion model, and we demonstrate its ability to efficiently improve low-quality plans by guiding the reduction of the estimated restoration gap through a process, which we call Restoration Gap Guidance (RGG). Furthermore, we propose a regularizer that prevents adversarial restoration gap guidance by utilizing an attribution map of the gap predictor. It effectively mitigates the risk of the plan being directed towards an unreliable plan, enabling further improvement in the planning performance.

The main contributions of this paper are summarized as follows: **(1)** We provide a novel metric to assess the quality of individual plans generated by the diffusion model with theoretical justification. **(2)** We propose a new generative process, Restoration Gap Guidance (RGG) which utilizes a gap predictor that estimates the restoration gap. **(3)** We show the effectiveness of our approach across three different benchmarks in offline control settings.

## 2 Background

### 2.1 Planning with Diffusion Probabilistic Models

We consider the reinforcement learning problem which aims to maximize the expected discounted sum of rewards $\mathbb{E}_\pi[\sum_{t=0}^T \gamma^t r(\boldsymbol{s}_t, \boldsymbol{a}_t)]$ where $\pi$ is a policy that defines a distribution over actions

$a_t$, $s_t$ represents the states that undergo transition according to unknown discrete-time dynamics $s_{t+1} = f(s_t, a_t)$, $r : \mathcal{S} \times \mathcal{A} \to \mathbb{R}$ is a reward function, and $\gamma \in (0, 1]$ is the discount factor. Trajectory optimization solves this problem by finding the sequence of actions $a_{0:T}^*$ that maximizes the expected discounted sum of rewards over planning horizon $T$:

$$a_{0:T}^* = \arg\max_{a_{0:T}} \mathcal{J}(\tau) = \arg\max_{a_{0:T}} \sum_{t=0}^{T} \gamma^t r(s_t, a_t), \tag{1}$$

where $\tau = (s_0, a_0, s_1, a_1, ..., s_t, a_t)$ represents a trajectory and $\mathcal{J}(\tau)$ denotes an objective value of that trajectory. This trajectory can be viewed as a particular form of two-dimensional sequence data:

$$\tau = \begin{bmatrix} s_0 & s_1 & & s_T \\ a_0 & a_1 & \cdots & a_T \end{bmatrix}. \tag{2}$$

Diffuser (Janner et al., 2022) is a trajectory planning model, which models a trajectory distribution by employing diffusion probabilistic models (Sohl-Dickstein et al., 2015; Ho et al., 2020):

$$p_\theta(\tau^0) = \int p(\tau^N) \prod_{i=1}^{N} p_\theta(\tau^{i-1}|\tau^i) \, d\tau^{1:N} \tag{3}$$

where $p(\tau^N)$ is a standard Gaussian prior, $\tau^0$ is a noiseless trajectory, and $p_\theta(\tau^{i-1}|\tau^i)$ is a denoising process which is a reverse of a forward process $q(\tau^i|\tau^{i-1})$ that gradually deteriorates the data structure by introducing noise. The denoising process is often parameterized as Gaussian with fixed timestep-dependent covariances: $p_\theta(\tau^{i-1}|\tau^i) = \mathcal{N}(\tau^{i-1}|\mu_\theta(\tau^i, i), \Sigma^i)$. Diffuser recasts the trajectory optimization problem as a conditional sampling with the conditional diffusion process under smoothness condition on $p(\mathcal{O}_{1:T} = 1|\tau)$ (Sohl-Dickstein et al., 2015):

$$\tilde{p}_\theta(\tau) = p(\tau|\mathcal{O}_{1:T} = 1) \propto p(\tau)p(\mathcal{O}_{1:T} = 1|\tau), \quad p_\theta(\tau^{i-1}|\tau^i, \mathcal{O}_{1:T}) \approx \mathcal{N}(\tau^{i-1}; \mu + \Sigma g, \Sigma) \tag{4}$$

where $\mu, \Sigma$ are the parameters of the denoising process $p_\theta(\tau^{i-1}|\tau^i)$, $\mathcal{O}_t$ is the optimality of timestep $t$ of trajectory with $p(\mathcal{O}_t = 1) = \exp(\gamma^t r(s_t, a_t))$ and

$$g = \nabla_\tau \log p(\mathcal{O}_{1:T}|\tau)|_{\tau=\mu} = \sum_{t=0}^{T} \gamma^t \nabla_{s_t, a_t} r(s_t, a_t)|_{(s_t, a_t) = \mu_t} = \nabla \mathcal{J}(\mu). \tag{5}$$

Therefore, a separate model $\mathcal{J}_\phi$ can be trained to predict the cumulative rewards of trajectory samples $\tau^i$. By utilizing the gradients of $\mathcal{J}_\phi$, trajectories with high cumulative rewards can be generated.

As part of the training procedure, Diffuser trains an $\epsilon$-model to predict the source noise instead of training $\mu_\theta$ as it turns out that learning $\epsilon_\theta$ enables the use of a simplified objective, where $\mu_\theta$ is easily recovered in a closed form (Ho et al., 2020):

$$\mathcal{L}(\theta) := \mathbb{E}_{i,\epsilon,\tau^0}[\|\epsilon - \epsilon_\theta(\tau^i, i)\|^2], \tag{6}$$

where $i \in \{0, 1, ..., N\}$ is the diffusion timestep, $\epsilon \sim \mathcal{N}(\mathbf{0}, \mathbf{I})$ is the target noise, and $\tau^i$ is the trajectory corrupted by the noise $\epsilon$ from the noiseless trajectory $\tau^0$.

## 2.2 Generalizing Diffusion Probabilistic Models as a Stochastic Differential Equation (SDE)

The forward process in diffusion probabilistic models perturbs data structure by gradually adding Gaussian noises. Under an infinite number of noise scales, this forward process over continuous time can be represented as a stochastic differential equation (SDE) (Song et al., 2021):

$$d\tau = \mathbf{f}(\tau, t) \, dt + g(t) \, d\mathbf{w}, \tag{7}$$

where $t \in (0, 1]$ is a continuous time variable for indexing diffusion timestep, $\mathbf{f}(\tau, t)$ is the drift coefficient, $g(t)$ is the diffusion coefficient, and $\mathbf{w}$ is the standard Wiener process. Similarly, the denoising process can be defined by the following reverse-time SDE:

$$d\tau = [\mathbf{f}(\tau, t) - g(t)^2 \mathbf{s}_\theta(\tau, t)] \, dt + g(t) \, d\bar{\mathbf{w}}, \tag{8}$$

where $\bar{\mathbf{w}}$ is the infinitesimal noise in the reverse time direction and $\mathbf{s}_\theta(\boldsymbol{\tau}, t)$ is the learned score network which estimates the data score $\nabla_{\boldsymbol{\tau}} \log p_t(\boldsymbol{\tau})$. This score network can be replaced by the $\epsilon$-model:

$$\mathbf{s}_\theta(\boldsymbol{\tau}, t) \approx \nabla_{\boldsymbol{\tau}} \log q(\boldsymbol{\tau}) = \mathbb{E}_{\boldsymbol{\tau}^\mathbf{o}}[\nabla_{\boldsymbol{\tau}} \log q(\boldsymbol{\tau}|\boldsymbol{\tau}^\mathbf{0})] = \mathbb{E}_{\boldsymbol{\tau}^\mathbf{o}}\left[-\frac{\boldsymbol{\epsilon}_\theta(\boldsymbol{\tau}, t)}{C_t}\right] = -\frac{\boldsymbol{\epsilon}_\theta(\boldsymbol{\tau}, t)}{C_t}, \quad (9)$$

where $C_t$ is a constant determined by the chosen perturbation strategies.

The solution of a forward SDE is a time-varying random variable $\boldsymbol{\tau}^t$. Using the reparameterization trick (Kingma & Welling, 2014), it is achieved by sampling a random noise $\boldsymbol{\epsilon}$ from a standard Gaussian distribution which is scaled by the target standard deviation $\sigma_t$ and shifted by the target mean:

$$\boldsymbol{\tau}^t = \alpha_t \boldsymbol{\tau}^0 + \sigma_t \boldsymbol{\epsilon}, \quad \boldsymbol{\epsilon} \sim \mathcal{N}(\mathbf{0}, \mathbf{I}), \quad (10)$$

where $\alpha_t : [0, 1] \to [0, 1]$ denotes a scalar function indicating the magnitude of the noiseless data $\boldsymbol{\tau}^0$, and $\sigma_t : [0, 1] \to [0, \infty)$ denotes a scalar function that determines the size of the noise $\boldsymbol{\epsilon}$. Depending on perturbation strategies for $\alpha_t$ and $\sigma_t$, two types of SDEs are commonly considered: the Variance Exploding SDE (VE-SDE) has $\alpha_t = 1$ for all $t$; whereas the Variance Preserving (VP) SDE satisfies $\alpha_t^2 + \sigma_t^2 = 1$ for all $t$. Both VE and VP SDE change the data distribution to random Gaussian noise as $t$ moves from 0 to 1. In this work, we describe diffusion probabilistic models within the continuous-time framework using VE-SDE to simplify notation, as VE/VP SDEs are mathematically equivalent under scale translations (Song et al., 2021).

For VE SDE, the forward process and denoising process are defined by the following SDEs:

$$\text{(Forward SDE)} \quad \mathrm{d}\boldsymbol{\tau} = \sqrt{\frac{\mathrm{d}[\sigma_t^2]}{\mathrm{d}t}} \, \mathrm{d}\mathbf{w} \quad (11)$$

$$\text{(Reverse SDE)} \quad \mathrm{d}\boldsymbol{\tau} = \left[-\frac{\mathrm{d}[\sigma_t^2]}{\mathrm{d}t}\mathbf{s}_\theta(\boldsymbol{\tau}, t)\right]\mathrm{d}t + \sqrt{\frac{\mathrm{d}[\sigma_t^2]}{\mathrm{d}t}} \, \mathrm{d}\bar{\mathbf{w}}. \quad (12)$$

## 3 Restoration Gap

To assess the quality of plans generated by diffusion probabilistic models, we propose a novel metric named *restoration gap*. It aims to automatically detect infeasible plans that violate system constraints. We hypothesize that for feasible plans, even if a certain amount of noise perturbs them, they can be closely restored to their initial plans by diffusion models. It is attributed to the property of temporal compositionality in diffusion planners (Janner et al., 2022) that encourages them to compose feasible trajectories by stitching together any feasible plan subsequences. However, for infeasible plans that obviously fall out of the training distribution as they violate physical constraints as shown in Figure 5, restoring them to a state near their original conditions is challenging. Based on this intuition, we define the restoration gap of the generated plan $\boldsymbol{\tau}$ as follows:

$$\text{perturb}_{\hat{t}}(\boldsymbol{\tau}) = \boldsymbol{\tau} + \sigma_{\hat{t}}\boldsymbol{\epsilon}_{\hat{t}}, \quad \boldsymbol{\epsilon}_{\hat{t}} \sim \mathcal{N}(\mathbf{0}, \mathbf{I}) \quad (13)$$

$$\text{restore}_{\hat{t},\theta}(\boldsymbol{\tau}) = \boldsymbol{\tau} + \int_{\hat{t}}^0 \left[-\frac{\mathrm{d}[\sigma_t^2]}{\mathrm{d}t}\mathbf{s}_\theta(\boldsymbol{\tau}, t)\right]\mathrm{d}t + \sqrt{\frac{\mathrm{d}[\sigma_t^2]}{\mathrm{d}t}} \, \mathrm{d}\bar{\mathbf{w}} \quad (14)$$

$$\text{restoration gap}_{\hat{t},\theta}(\boldsymbol{\tau}) = \mathbb{E}_{\boldsymbol{\epsilon}_{\hat{t}}}\left[\|\boldsymbol{\tau} - \text{restore}_{\hat{t},\theta}(\text{perturb}_{\hat{t}}(\boldsymbol{\tau}))\|_2\right], \quad (15)$$

where $\hat{t} \in (0, 1]$ indicates the magnitude of applied perturbation. The restoration gap measures the expected $L_2$ distance between the generated plan and the plan restored from noise corruption, which is estimated by the Monte Carlo approximation.

Figure 2 provides empirical evidence supporting our hypothesis. In order to analyze the effectiveness of the restoration gap, we define artifact plans generated by Diffuser (Janner et al., 2022) that involve transitions of passing through walls for which it is impossible for the agent to follow. We compare the distribution of the restoration gap for both groups, normal plans and artifact plans[1]. The histogram of the restoration gap for normal and artifact plans demonstrates that infeasible artifact plans have larger

---

[1]The purpose of defining artifacts in this manner is solely to validate our hypothesis. Artifacts are not explicitly defined beyond the scope of this validation.

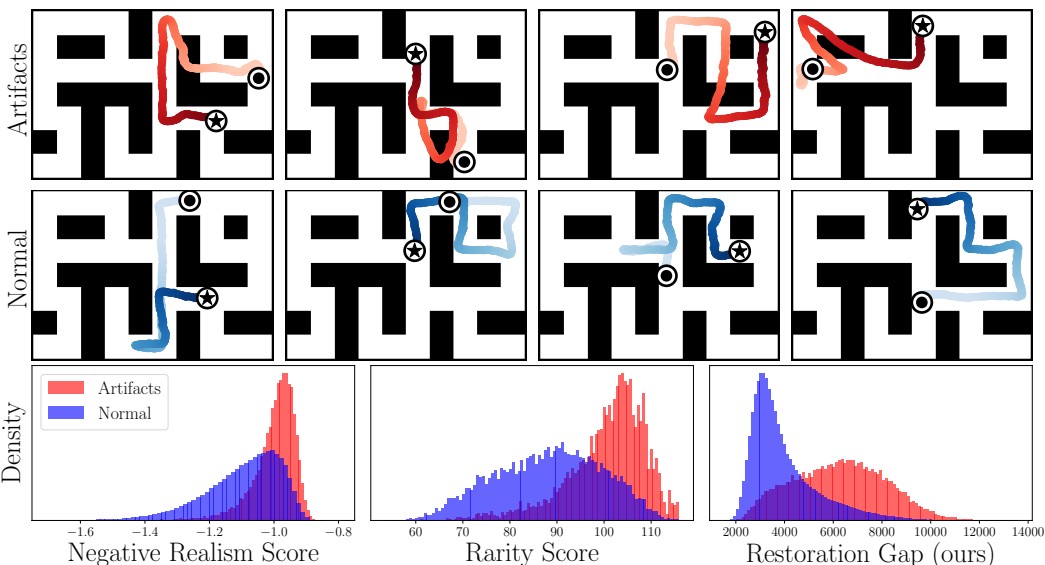

Figure 2: The first and second rows show examples of artifact and normal plans, respectively, generated by Diffuser (Janner et al., 2022) in the Maze2D-Large environment, including a predetermined start ⊙ and goal ★. The third row presents the density of realism score (Kynkäänniemi et al., 2019), rarity score (Han et al., 2023), and restoration gap to illustrate the differences in distribution between artifacts and normal plans. Detailed explanation of other metrics is described in Appendix C.2.

restoration gap values compared to normal plans. Therefore, the detection of infeasible artifact plans can be automated by incorporating a statistical test that utilizes the restoration gap and thresholding with a threshold value of $b > 0$:

$$\text{restoration gap}_{\hat{t},\theta}(\boldsymbol{\tau}) > b. \tag{16}$$

To bound the probability of making errors by choosing the specific threshold $b$, we provide Proposition 1. Let $\mathbb{H}_0$ represent the null hypothesis which assumes that the trajectory $\boldsymbol{\tau}$ belongs to the normal set $\mathcal{T}_{\text{normal}}$, and let $\mathbb{H}_1$ represent the alternative hypothesis which assumes that the trajectory $\boldsymbol{\tau}$ belongs to the artifact set $\mathcal{T}_{\text{artifacts}}$. The following proposition suggests how to choose the threshold $b$ in order to bound the error probabilities.

**Proposition 1.** *Given $t \in [0,1]$ and a positive constant $C, \Delta$, assume that $\|\mathbf{s}_\theta(\boldsymbol{\tau},t)\|_2^2 \leq C^2$ for all $\boldsymbol{\tau} \in \mathcal{T}_{\text{normal}} \subset \mathbb{R}^d$, and $\|\mathbf{s}_\theta(\boldsymbol{\tau},t)\|_2^2 \geq (C+\Delta)^2$ for all $\boldsymbol{\tau} \in \mathcal{T}_{\text{artifacts}} \subset \mathbb{R}^d$. If*

$$\Delta \geq \frac{2\sqrt{d} + 2\sqrt{d + 2\sqrt{-d \cdot \log \delta} - 2 \log \delta}}{\sigma_{\hat{t}}}, \tag{17}$$

*then setting*

$$b \geq \sigma_{\hat{t}} \left( C \sigma_{\hat{t}} + \sqrt{d} + \sqrt{d + 2\sqrt{-d \cdot \log \delta} - 2 \log \delta} \right) \tag{18}$$

*guarantees both type I and type II errors at most $2\delta$.*

*Proof Sketch.* We begin by deriving thresholds $b_I$ and $b_{II}$ to control type I and type II errors at most $\delta$, respectively. This is done by decomposing the restoration gap into the outcomes of the score and Gaussian noise. To ensure the control of both type I and type II errors, we examine the condition $b_I \leq b_{II}$ and obtain the conclusion. For the complete proof, see Appendix A. □

According to Proposition 1, to achieve low error probabilities for both type I (false positives, where normal trajectories are incorrectly classified as artifacts) and type II (false negatives, where artifact trajectories are wrongly identified as normal) errors, it is essential to have a large enough $\sigma_{\hat{t}}$ to properly satisfy the condition, which implies having a large enough $\hat{t}$. In practice, we find that setting $\hat{t} = 0.9$ works well.

# 4 Refining Diffusion Planner

## 4.1 Restoration Gap Guidance

Although Diffuser (Janner et al., 2022) has demonstrated competitive performance against previous non-diffusion-based planning methods by utilizing gradients of return $\mathcal{J}_\phi$ to guide trajectories during the denoising process:

$$\mathrm{d}\boldsymbol{\tau} = [\mathbf{f}(\boldsymbol{\tau}, t) - g(t)^2 (\mathbf{s}_\theta(\boldsymbol{\tau}, t) + \alpha \nabla \mathcal{J}_\phi(\boldsymbol{\tau}))] \, \mathrm{d}t + g(t) \, \mathrm{d}\bar{\mathbf{w}}, \tag{19}$$

it entirely relies on the ability of a generative model and assumes a perfect data score estimation. For plans with inaccurately estimated scores, the diffusion models could generate unreliable plans that are infeasible to execute and lead to limited performance. To address this, it is essential to construct an adjusted score to refine the generative process of the diffusion planner. Therefore, we estimate the restoration gap by training a gap predictor $\mathcal{G}_\psi$ on synthetic diffused data generated through the diffusion process, taking full advantage of its superior generation ability with conditional guidance from gradients of return. Parameters of the gap predictor $\psi$ are optimized by minimizing the following objective:

$$\mathcal{L}(\psi) := \mathbb{E}_{t, \boldsymbol{\tau}^0}[\|\text{restoration gap}_{\hat{t}, \theta}(\boldsymbol{\tau}^t) - \mathcal{G}_\psi(\boldsymbol{\tau}^t, t)\|^2], \tag{20}$$

where $t \in (0, 1]$ denotes a continuous time variable for indexing the diffusion timestep, and $\boldsymbol{\tau}^t$ is the diffused trajectory resulting from $\boldsymbol{\tau}^0$ at diffusion timestep $t$. With this gap predictor, we define the Restoration Gap Guidance (RGG) as follows:

$$\mathrm{d}\boldsymbol{\tau} = [\mathbf{f}(\boldsymbol{\tau}, t) - g(t)^2 (\mathbf{s}_\theta(\boldsymbol{\tau}, t) + \alpha(\nabla \mathcal{J}_\phi(\boldsymbol{\tau}) - \beta \nabla \mathcal{G}_\psi(\boldsymbol{\tau}, t)))] \, \mathrm{d}t + g(t) \, \mathrm{d}\bar{\mathbf{w}}, \tag{21}$$

where $\alpha$ is a positive coefficient that scales the overall guidance and $\beta$ is a positive coefficient that can be adjusted to enforce a small restoration gap for the generated trajectory.

## 4.2 Attribution Map Regularization

Although guiding the diffusion planner to minimize the restoration gap effectively refines low-quality plans (more details in Section 5), this refining guidance could push the plan in an undesirable direction due to the estimation error of the gap predictor during the denoising process. As a result of this estimation error, guiding plans with the sub-optimal gap predictor may result in *model exploitation* (Kurutach et al., 2018; Janner et al., 2019; Rajeswaran et al., 2020), yielding sub-optimal results.

To mitigate the issue of adversarial guidance, we present a regularization method that prevents the gap predictor from directing plans in the wrong direction. Inspired by the prior studies which improve the model performance by utilizing attribution maps (Nagisetty et al., 2020; Bertoin et al., 2022), we measure a total variation of the attribution map $M$ obtained from any input attribution methods $M = E(\mathcal{G}_\psi(\boldsymbol{\tau}, t))$. Each element of the attribution map indicates the extent to which the final prediction is influenced by the corresponding input feature. The rationale of employing the total variation of $M$ lies in the

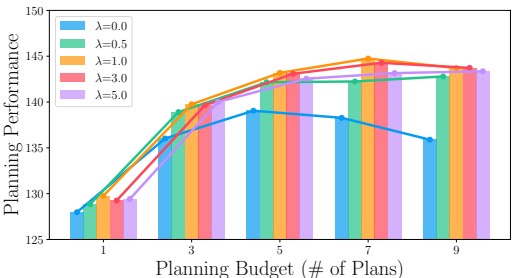

Figure 3: Planning performance of RGG+ on Maze2D-Large single-task with varying $\lambda$ values.

hypothesis that transitions with excessively high attribution scores are more likely to be outliers. This is because a sequence of transitions within a planned trajectory, rather than a single one, causes a plan to have a high restoration gap. By adding this attribution map regularization, Equation 21 becomes:

$$\mathrm{d}\boldsymbol{\tau} = [\mathbf{f}(\boldsymbol{\tau}, t) - g(t)^2 (\mathbf{s}_\theta(\boldsymbol{\tau}, t) + \alpha(\nabla \mathcal{J}_\phi(\boldsymbol{\tau}) - \beta \nabla \mathcal{G}_\psi(\boldsymbol{\tau}, t) - \lambda \nabla \|\nabla M\|))] \, \mathrm{d}t + g(t) \, \mathrm{d}\bar{\mathbf{w}}, \tag{22}$$

where $\lambda$ is a control parameter given by a positive constant, encouraging the attribution map to have a simple, organized structure while preventing the occurrence of adversarial artifacts. We refer to this modification as RGG+.

Table 1: The performance of RGG, RGG+, and various previous algorithms, measured as normalized average return, is presented on the D4RL locomotion benchmark (Fu et al., 2020). Results for RGG and RGG+ show the mean and standard error over 15 planning seeds. Detailed sources for the performance of prior methods are provided in Appendix E.

| Dataset | Environment | BC | CQL | IQL | DT | TT | MOPO | MOReL | MBOP | Diffuser | RGG | RGG+ |
|---------|-------------|----|-----|-----|----|----|------|-------|------|----------|-----|------|
| Med-Expert | HalfCheetah | 55.2 | 91.6 | 86.7 | 86.8 | 95.0 | 63.3 | 53.3 | 105.9 | 79.8 | 90.8 ± 0.3 | 91.2 ± 0.3 |
| Med-Expert | Hopper | 52.5 | 105.4 | 91.5 | 107.6 | 110.0 | 23.7 | 108.7 | 55.1 | 107.2 | 109.6 ± 2.3 | 109.9 ± 2.3 |
| Med-Expert | Walker2d | 107.5 | 108.8 | 109.6 | 108.1 | 101.9 | 44.6 | 95.6 | 70.2 | 108.4 | 107.8 ± 0.1 | 107.7 ± 0.2 |
| Medium | HalfCheetah | 42.6 | 44.0 | 47.4 | 42.6 | 46.9 | 42.3 | 42.1 | 44.6 | 44.2 | 44.0 ± 0.3 | 44.2 ± 0.3 |
| Medium | Hopper | 52.9 | 58.5 | 66.3 | 67.6 | 61.1 | 28.0 | 95.4 | 48.8 | 58.5 | 82.5 ± 4.3 | 84.9 ± 4.1 |
| Medium | Walker2d | 75.3 | 72.5 | 78.3 | 74.0 | 79.0 | 17.8 | 77.8 | 41.0 | 79.7 | 81.7 ± 0.5 | 82.0 ± 0.4 |
| Med-Replay | HalfCheetah | 36.6 | 45.5 | 44.2 | 36.6 | 41.9 | 53.1 | 40.2 | 42.3 | 42.2 | 41.0 ± 0.2 | 41.3 ± 0.2 |
| Med-Replay | Hopper | 18.1 | 95.0 | 94.7 | 82.7 | 91.5 | 67.5 | 93.6 | 12.4 | 96.8 | 95.2 ± 0.5 | 95.8 ± 0.5 |
| Med-Replay | Walker2d | 26.0 | 77.2 | 73.9 | 66.6 | 82.6 | 39.0 | 49.8 | 9.7 | 61.2 | 78.3 ± 4.4 | 77.5 ± 4.7 |
| **Average** | | 51.9 | 77.6 | 77.0 | 74.7 | 78.9 | 42.1 | 72.9 | 47.8 | 75.3 | **81.2** | **81.6** |

# 5    Experiments

We present the analytical results of approaches to improve planning performance by leveraging guidance from our proposed metric, the restoration gap, for a wide range of decision-making tasks in offline control settings. Specifically, we demonstrate **(1)** the relationship between a high restoration gap and poor planning performance, **(2)** the enhancement of planning performance in the diffusion planner by leveraging restoration gap guidance, and **(3)** explainability by presenting the attribution maps of the learned gap predictor, highlighting infeasible transitions. More information about our experimental setup and implementation details can be found in Appendix C and Appendix D, respectively.

## 5.1    Relationship between Restoration Gap and Planning Performance

We evaluate how effectively the restoration gap can identify infeasible plans by comparing our metric with a realism score (Kynkäänniemi et al., 2019) and rarity score (Han et al., 2023). Both prior metrics are designed to assess the quality of generated samples by examining the discrepancy between the generated sample and the real data manifold in the feature space. Figure 4 illustrates the performance of plans which are chosen up to top-K% from each metric. As illustrated in Figure 4, the higher the restoration gap of the plan is, the poorer the performance is, which implies that the restoration gap captures the quality of the plan well compared to other metrics.

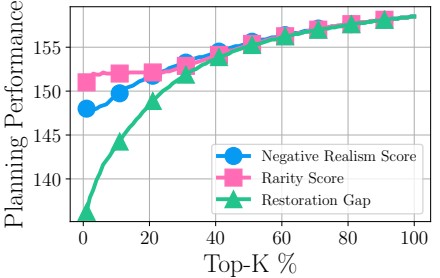

Figure 4: Performance of plans chosen from the top-K % considering various metrics, in the Maze2D-Large single-task.

## 5.2    Planning Performance Enhancement

**Maze2D Experiments**   Maze2D environments (Fu et al., 2020) involve a navigation task that requires an agent to exhibit long-horizon planning abilities to reach a target goal location. Maze2D environments consist of two tasks: a single-task where the goal location is fixed, and a multi-task which we refer to as Multi2D where the goal location is randomized at the beginning of every episode. We compare our methods

Table 2: Diffuser with RGG and Diffuser with RGG+ outperform all baselines. We report the mean and the standard error over 1000 planning seeds.

| Environment | | CQL | MPPI | IQL | Diffuser | RGG | RGG+ |
|-------------|--|-----|------|-----|----------|-----|------|
| Maze2D | U-Maze | 5.7 | 33.2 | 47.4 | 108.6 ± 1.4 | 108.8 ± 1.4 | 109.5 ± 1.3 |
| Maze2D | Medium | 5.0 | 10.2 | 34.9 | 129.8 ± 0.7 | 131.8 ± 0.5 | 132.1 ± 0.4 |
| Maze2D | Large | 12.5 | 5.1 | 58.6 | 123.5 ± 2.0 | 135.4 ± 1.7 | 143.9 ± 1.5 |
| **Single-task Average** | | 7.7 | 16.2 | 47.0 | 120.6 | **125.3** | **128.5** |
| Multi2D | U-Maze | - | 41.2 | 24.8 | 127.9 ± 0.8 | 128.3 ± 0.8 | 128.3 ± 0.8 |
| Multi2D | Medium | - | 15.4 | 12.1 | 130.1 ± 0.9 | 130.0 ± 0.9 | 130.0 ± 0.9 |
| Multi2D | Large | - | 8.0 | 13.9 | 141.2 ± 1.6 | 148.3 ± 1.4 | 150.9 ± 1.3 |
| **Multi-task Average** | | - | 21.5 | 16.9 | 133.1 | **135.5** | **136.4** |

with the model-free offline reinforcement learning algorithms CQL (Kumar et al., 2020) and IQL (Kostrikov et al., 2022); conventional trajectory optimizer MPPI (Williams et al., 2015); and sequence modeling approach Diffuser (Janner et al., 2022). As shown in Table 2, RGG improves the planning performance of Diffuser in 5 out of 6 tasks, with notable improvements in the Maze2D-Large envi-

ronments where the complexity of the obstacle maps is higher than in U-Maze or Medium layouts, leading to a higher occurrence of infeasible plans. RGG+ performs on par with or better than RGG. In contrast, model-free algorithms fail to reliably achieve the goal, as Maze2D environments require hundreds of steps to arrive at the goal location.

**Locomotion Experiments**    Gym-MuJoCo locomotion tasks (Fu et al., 2020) are standard benchmarks in evaluating algorithms on heterogeneous data with varying quality. We compare our methods with the model-free algorithms CQL (Kumar et al., 2020) and IQL (Kostrikov et al., 2022); model-based algorithms MOPO (Yu et al., 2020), MOReL (Kidambi et al., 2020), and MBOP (Argenson & Dulac-Arnold, 2021); sequence modeling approach Decision Transformer (DT) (Chen et al., 2021), Trajectory Transformer (TT) (Janner et al., 2021) and Diffuser (Janner et al., 2022); and pure imitation-based approach behavior-cloning (BC). As indicated in Table 1, our approach of refining Diffuser with RGG either matches or surpasses most of the offline RL baselines when considering the average score across various tasks. Additionally, it significantly enhances the performance of Diffuser, particularly in the "Medium" dataset. We attribute this improvement to the sub-optimal and exploratory nature of the policy that was used to generate the "Medium" dataset, which results in a challenging data distribution to learn the diffusion planner. Consequently, RGG clearly contributes to the enhancement of planning performance. However, RGG+ only brings about a marginal improvement over RGG. This might be because we adopt the strategy of (Janner et al., 2022), using a closed-loop controller and a shorter planning horizon in locomotion environments compared to Maze2D environments, thereby simplifying the learning process of the gap predictor.

**Block Stacking Experiments**    The block stacking task suite with a Kuka iiwa robotic arm is a benchmark to evaluate the model performance for a large state space (Janner et al., 2022) where the offline demonstration data is achieved by PDDLStream (Garrett et al., 2020). It involves two tasks: an unconditional stacking task whose goal is to

Table 3: The performance of RGG, RGG+, and various prior methods evaluated over 100 planning seeds. A score of 100 is desired, while a random approach would receive a score of 0.

| Environment | BCQ | CQL | Diffuser | RGG | RGG+ |
|---|---|---|---|---|---|
| Unconditional Stacking | 0.0 | 24.4 | 53.3 ± 2.4 | 63.3 ± 2.7 | 65.3 ± 2.0 |
| Conditional Stacking | 0.0 | 0.0 | 44.3 ± 3.2 | 53.0 ± 3.3 | 56.7 ± 3.1 |
| **Average** | 0.0 | 8.1 | 48.8 | **58.2** | **61.0** |

maximize the height of a block tower, and a conditional stacking task whose goal is to stack towers of blocks subject to a specified order of blocks. We compare our methods with model-free offline reinforcement learning algorithms BCQ (Fujimoto et al., 2019) and CQL (Kumar et al., 2020), and Diffuser (Janner et al., 2022). We present quantitative results in Table 3, where a score of 100 corresponds to the successful completion of the task. The results demonstrate the superior performance of RGG over all baselines, with RGG+ further enhancing this planning performance.

### 5.3    Injecting Explainability to Diffusion Planners

The explainability of decision-making models is particularly important in control domains as they could potentially harm physical objects including humans (Kim & Choi, 2021; Lee et al., 2023; Beechey et al., 2023; Kim et al., 2023; Kenny et al., 2023). Training the gap predictor enables the diffusion planner to have explainability. Diffusion planners often generate trajectories with unreliable transitions resulting in execution failures. Attribution maps from the gap predictor highlight such unreliable transitions by identifying the extent to which each transition contributes to the decision of the gap predictor. Specifically, in Maze2D, the attribution maps emphasize the transitions involving wall-crossing or abrupt directional changes, as illustrated in Figure 5. In the unconditional block stacking task where the robot destroys the tower while stacking the last block, the tower-breaking transitions are highlighted. On the other hand, for successful trajectories on the second and third attribution maps, the attribution maps do not emphasize picking or stacking behaviors. Similarly, in the conditional block stacking task where the robot fails to stack the block, they spotlight the transitions of stacking behaviors.

### 5.4    Additional Experiments

To study the benefit of regularization on harder tasks, characterized by a larger trajectory space and a smaller fraction of the space observed in training, we explore $\lambda$ values $[0.0, 0.5, 1.0, 3.0, 5.0]$ while increasing the planning budget as illustrated in Figure 3. As the planning budget increases, $\lambda = 0$ generates adversarial plans, resulting in decreased performance. In contrast, RGG+ demonstrates

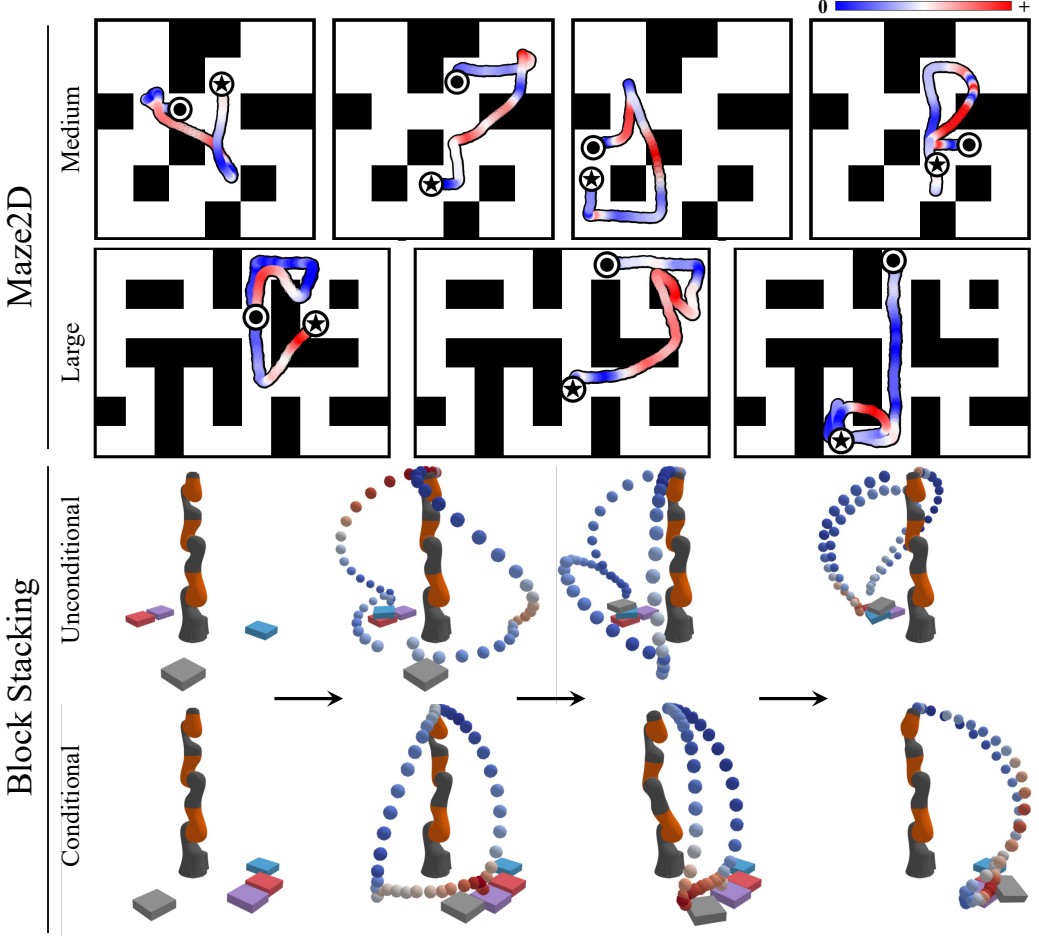

Figure 5: Attribution maps for trajectories generated by diffusion planner highlight transitions that have a substantial contribution to the estimation of a high restoration gap by the gap predictor, indicated in red.

effectiveness across a wide range of $\lambda$ values, with $\lambda > 0$ consistently outperforming $\lambda = 0$ (i.e., better than no regularization). Further investigation into the attribution method, perturbation magnitude, and comparison with guidance approaches, including metrics such as the rarity score, the negative realism score, and the discriminator, as well as the visualization of low and high restoration gap plans, can be found in Appendix B.

## 6   Related Work

**Metrics for Evaluating Generative Model**     Inception score (IS) (Salimans et al., 2016) and Fréchet inception distance (FID) (Heusel et al., 2017) are commonly used as standard evaluation metrics for generative models, assessing the quality of generated samples by comparing the discrepancy between real and generated samples in the feature space. However, these metrics do not distinguish between fidelity and diversity aspects of generated samples. To address this issue, precision and recall variants (Sajjadi et al., 2018; Kynkäänniemi et al., 2019) are introduced to separately evaluate these properties. Subsequently, density and coverage (Naeem et al., 2020) are proposed to overcome some of the drawbacks of precision and recall, such as vulnerability to outliers and computational inefficiency. While these metrics are helpful for evaluating the quality of a set of generated samples, they are not suitable for ranking individual samples. In contrast, realism score (Kynkäänniemi et al., 2019) and rarity score (Han et al., 2023) offer a continuous extension of improved precision and recall, enabling the evaluation of individually generated sample quality. Despite their usefulness,

these methods come with limitations as they rely on real samples for precise real manifold estimation, whereas our restoration gap does not have such a constraint.

**Diffusion Model in Reinforcement Learning**   Diffusion models have gained prominence as a notable class of generative models, characterizing the data generation process through iterative denoising procedure (Sohl-Dickstein et al., 2015; Ho et al., 2020). This denoising procedure can be viewed as a way to parameterize the gradients of the data distribution (Song & Ermon, 2019), linking diffusion models to score matching (Hyvärinen & Dayan, 2005) and energy-based models (EBMs) (LeCun et al., 2006; Du & Mordatch, 2019; Nijkamp et al., 2019; Grathwohl et al., 2020). Recently, diffusion models have been successfully applied to various control tasks (Janner et al., 2022; Urain et al., 2023; Ajay et al., 2023; Chi et al., 2023; Liang et al., 2023). In particular, Diffuser (Janner et al., 2022) employs an unconditional diffusion model to generate trajectories consisting of state-action pairs. The approach includes training a separate model that predicts the cumulative rewards of noisy trajectory samples, which then guides the reverse diffusion process towards high-return trajectory samples in the inference phase, analogous to classifier-guided sampling (Dhariwal & Nichol, 2021). Building upon this, Decision Diffuser (Ajay et al., 2023) extends the capabilities of Diffuser by adopting a conditional diffusion model with reward or constraint guidance to effectively satisfy constraints, compose skills, and maximize return. Meanwhile, AdaptDiffuser (Liang et al., 2023) enhances generalization ability of the diffusion model to unseen tasks by selectively fine-tuning it with high-quality data, derived through the use of hand-designed reward functions and an inverse dynamics model. In contrast, in this work, we focus on evaluating the quality of individually generated samples and explore ways to enhance planning performance by utilizing guidance derived from these evaluations.

**Restoring Artifacts in Generative Models**   Recently, several studies have concentrated on investigating the artifacts in Generative Adversarial Networks (GAN) model architectures for image generation tasks. GAN Dissection (Bau et al., 2019) explores the internal mechanisms of GANs, focusing on the identification and removal of units that contribute to artifact production, leading to more realistic outputs. In a subsequent study, an external classifier is trained to identify regions of low visual fidelity in individual generations and to detect internal units associated with those regions (Tousi et al., 2021). Alternatively, artifact correction through latent code manipulation based on a binary linear classifier is proposed (Shen et al., 2020). Although these methods can assess the fidelity of individual samples, they still necessitate additional supervision, such as human annotation. To address this limitation, subsequent works explore unsupervised approaches for detecting and correcting artifact generations by examining local activation (Jeong et al., 2022) and activation frequency (Choi et al., 2022). In contrast, our work primarily focuses on refining the generative process of diffusion probabilistic models to restore low-quality plans.

# 7   Conclusion

We have presented a novel refining method that fixes infeasible transitions within the trajectory generated by the diffusion planner. This refining process is guided by a proposed metric, restoration gap, which quantifies the restorability of a given plan. Under specific regularity conditions, we prove that the restoration gap effectively identifies unreliable plans while ensuring a low error probability for both type I and type II errors. The experimental results, which include enhancement in quantitative planning performance and visualization of qualitative attribution maps, highlight the importance of the refinement method of the diffusion planner.

**Limitations**   While the restoration gap guidance effectively enhances the feasibility of plans and consistently improves the planning performance of diffusion models, our method is limited in situations where an offline dataset is provided. Training the diffusion model often requires transition data that uniformly covers the state-action space, the collection of which is a nontrivial and time-consuming task.

**Future Work**   Our analysis of the effectiveness of the restoration gap is currently confined to a relatively simple task, Maze2D (see Figure 2), where we explicitly define normal and artifact plans. The choice of Maze2D is motivated by its suitability for identifying violations of prior knowledge, such as feasible plans not passing through walls. However, as future work, it would be worthwhile to explore the efficacy of restoration gap in more complex tasks, such as the block stacking task.

## Acknowledgements

This work was supported by the Industry Core Technology Development Project, 20005062, Development of Artificial Intelligence Robot Autonomous Navigation Technology for Agile Movement in Crowded Space, funded by the Ministry of Trade, Industry & Energy (MOTIE, Republic of Korea) and by Institute of Information & communications Technology Planning & Evaluation (IITP) grant funded by the Korea government (MSIT) (No. 2022-0-00984, Development of Artificial Intelligence Technology for Personalized Plug-and-Play Explanation and Verification of Explanation, No.2019-0-00075, Artificial Intelligence Graduate School Program (KAIST)).

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

# Appendix A  Proofs

**Proposition 1.** *Given $t \in [0,1]$ and a positive constant $C, \Delta$, assume that $\|\mathbf{s}_\theta(\boldsymbol{\tau}, t)\|_2^2 \le C^2$ for all $\boldsymbol{\tau} \in \mathcal{T}_{\text{normal}} \subset \mathbb{R}^d$, and $\|\mathbf{s}_\theta(\boldsymbol{\tau}, t)\|_2^2 \ge (C + \Delta)^2$ for all $\boldsymbol{\tau} \in \mathcal{T}_{\text{artifacts}} \subset \mathbb{R}^d$. If*

$$\Delta \ge \frac{2\sqrt{d} + 2\sqrt{d + 2\sqrt{-d \cdot \log \delta} - 2 \log \delta}}{\sigma_{\hat{t}}}, \tag{17}$$

*then setting*

$$b \ge \sigma_{\hat{t}} \left( C\sigma_{\hat{t}} + \sqrt{d} + \sqrt{d + 2\sqrt{-d \cdot \log \delta} - 2 \log \delta} \right) \tag{18}$$

*guarantees both type I and type II errors at most $2\delta$.*

*Proof.* We prove the proposition by employing techniques similar to those used in Theorem 3.3 from (Han et al., 2019) and Proposition 1 from (Meng et al., 2022). To guarantee that both Type I and Type II errors are at most $2\delta$, we first derive thresholds $b_I$ and $b_{II}$ to control Type I and Type II errors at most $\delta$, respectively.

**Controlling Type I Error**

For the type I error, we aim to bound the probability:

$$\mathbb{P}(\text{restoration gap}_{\hat{t}, \theta}(\boldsymbol{\tau}) \ge b_I | \mathbb{H}_0), \tag{23}$$

where $b_I$ represents the acceptance threshold for the alternative hypothesis. The restoration gap$_{\hat{t}, \theta}(\boldsymbol{\tau})$ can be written as

$$\text{restoration gap}_{\hat{t}, \theta}(\boldsymbol{\tau}) \tag{24}$$

$$= \mathbb{E}_{\epsilon_{\hat{t}}} \left[ \left\| \boldsymbol{\tau} - \text{restore}_{\hat{t}, \theta}(\text{perturb}_{\hat{t}}(\boldsymbol{\tau})) \right\|_2 \right] \tag{25}$$

$$= \mathbb{E}_{\epsilon_{\hat{t}}} \left[ \left\| \boldsymbol{\tau} - \left( \boldsymbol{\tau} + \sigma_{\hat{t}} \epsilon_{\hat{t}} + \int_{\hat{t}}^0 \left[ -\frac{\mathrm{d}[\sigma_t^2]}{\mathrm{d}t} \mathbf{s}_\theta(\boldsymbol{\tau}, t) \right] \mathrm{d}t + \sqrt{\frac{\mathrm{d}[\sigma_t^2]}{\mathrm{d}t}} \, \mathrm{d}\bar{\mathbf{w}} \right) \right\|_2 \right] \tag{26}$$

$$= \mathbb{E}_{\epsilon_{\hat{t}}} \left[ \left\| \sigma_{\hat{t}} \epsilon_{\hat{t}} + \int_{\hat{t}}^0 \left[ -\frac{\mathrm{d}[\sigma_t^2]}{\mathrm{d}t} \mathbf{s}_\theta(\boldsymbol{\tau}, t) \right] \mathrm{d}t + \sqrt{\frac{\mathrm{d}[\sigma_t^2]}{\mathrm{d}t}} \, \mathrm{d}\bar{\mathbf{w}} \right\|_2 \right] \tag{27}$$

$$\le \sigma_{\hat{t}} \mathbb{E}_{\epsilon_{\hat{t}}} \left[ \|\epsilon_{\hat{t}}\|_2 \right] + \left\| \int_{\hat{t}}^0 \left[ -\frac{\mathrm{d}[\sigma_t^2]}{\mathrm{d}t} \mathbf{s}_\theta(\boldsymbol{\tau}, t) \right] \mathrm{d}t + \sqrt{\frac{\mathrm{d}[\sigma_t^2]}{\mathrm{d}t}} \, \mathrm{d}\bar{\mathbf{w}} \right\|_2 \tag{28}$$

$$\le \sigma_{\hat{t}} \mathbb{E}_{\epsilon_{\hat{t}}} \left[ \|\epsilon_{\hat{t}}\|_2 \right] + \left\| \int_{\hat{t}}^0 \left[ -\frac{\mathrm{d}[\sigma_t^2]}{\mathrm{d}t} \mathbf{s}_\theta(\boldsymbol{\tau}, t) \right] \mathrm{d}t \right\|_2 + \left\| \int_{\hat{t}}^0 \sqrt{\frac{\mathrm{d}[\sigma_t^2]}{\mathrm{d}t}} \, \mathrm{d}\bar{\mathbf{w}} \right\|_2 \tag{29}$$

$$\le \sqrt{d} \sigma_{\hat{t}} + C\sigma_{\hat{t}}^2 + \left\| \int_{\hat{t}}^0 \sqrt{\frac{\mathrm{d}[\sigma_t^2]}{\mathrm{d}t}} \, \mathrm{d}\bar{\mathbf{w}} \right\|_2, \tag{30}$$

where the last inequality comes from Jensen's inequality, $\mathbb{E}_{\epsilon_{\hat{t}}} \left[ \|\epsilon_{\hat{t}}\|_2 \right] \le \sqrt{\mathbb{E}_{\epsilon_{\hat{t}}} \left[ \|\epsilon_{\hat{t}}\|_2^2 \right]} = \sqrt{d}$, and considering the assumption over $\mathbf{s}_\theta(\boldsymbol{\tau}, t)$ under the null hypothesis. As shown in (Meng et al., 2022), the last term corresponds to the $L_2$ norm of a random variable arising from a Wiener process at time $t = 0$, where its marginal distribution is given by $\epsilon \sim \mathcal{N}(\mathbf{0}, \sigma_{\hat{t}}^2 \mathbf{I})$. Dividing the squared $L_2$ norm of $\epsilon$ by $\sigma_t^2$ results in a $\chi^2$-distribution with $d$ degrees of freedom. According to Lemma 1 from (Laurent & Massart, 2000), we obtain the following one-sided tail bound:

$$\mathbb{P}(\|\epsilon\|_2^2 / \sigma_{\hat{t}}^2 \ge d + 2\sqrt{-d \cdot \log \delta} - 2 \log \delta) \le \exp(\log \delta) = \delta. \tag{31}$$

Then, we have,

$$\mathbb{P} \left( \|\epsilon\|_2 / \sigma_{\hat{t}} \ge \sqrt{d + 2\sqrt{-d \cdot \log \delta} - 2 \log \delta} \right) \le \delta. \tag{32}$$

Therefore, under null hypothesis, with probability of at least $(1 - \delta)$, we have that:

$$\text{restoration gap}_{\hat{t},\theta}(\boldsymbol{\tau}) \leq b_I = \sigma_t \left( C\sigma_t + \sqrt{d} + \sqrt{d + 2\sqrt{-d \cdot \log \delta} - 2\log \delta} \right) \tag{33}$$

which guarantees a bounded type I error at most $\delta$.

**Controlling Type II Error**

For the type II error, we aim to bound the probability:

$$\mathbb{P}(\text{restoration gap}_{\hat{t},\theta}(\boldsymbol{\tau}) \leq b_{II}|\mathbb{H}_1), \tag{34}$$

where $b_{II}$ represents the acceptance threshold for the alternative hypothesis. The restoration $\text{gap}_{\hat{t},\theta}(\boldsymbol{\tau})$ can be written as

$$\text{restoration gap}_{\hat{t},\theta}(\boldsymbol{\tau}) \tag{35}$$

$$= \mathbb{E}_{\epsilon_{\hat{t}}} \left[ \left\| \boldsymbol{\tau} - \text{restore}_{\hat{t},\theta}(\text{perturb}_{\hat{t}}(\boldsymbol{\tau})) \right\|_2 \right] \tag{36}$$

$$= \mathbb{E}_{\epsilon_{\hat{t}}} \left[ \left\| \boldsymbol{\tau} - (\boldsymbol{\tau} + \sigma_{\hat{t}}\epsilon_{\hat{t}} + \int_{\hat{t}}^0 \left[ -\frac{\mathrm{d}[\sigma_t^2]}{\mathrm{d}t} \mathbf{s}_\theta(\boldsymbol{\tau}, t) \right] \mathrm{d}t + \sqrt{\frac{\mathrm{d}[\sigma_t^2]}{\mathrm{d}t}} \, \mathrm{d}\bar{\mathbf{w}}) \right\|_2 \right] \tag{37}$$

$$= \mathbb{E}_{\epsilon_{\hat{t}}} \left[ \left\| \sigma_{\hat{t}}\epsilon_{\hat{t}} + \int_{\hat{t}}^0 \left[ -\frac{\mathrm{d}[\sigma_t^2]}{\mathrm{d}t} \mathbf{s}_\theta(\boldsymbol{\tau}, t) \right] \mathrm{d}t + \sqrt{\frac{\mathrm{d}[\sigma_t^2]}{\mathrm{d}t}} \, \mathrm{d}\bar{\mathbf{w}} \right\|_2 \right] \tag{38}$$

$$\geq -\sigma_{\hat{t}}\mathbb{E}_{\epsilon_{\hat{t}}}\left[\|\epsilon_{\hat{t}}\|_2\right] + \left\| \int_{\hat{t}}^0 \left[ -\frac{\mathrm{d}[\sigma_t^2]}{\mathrm{d}t} \mathbf{s}_\theta(\boldsymbol{\tau}, t) \right] \mathrm{d}t + \sqrt{\frac{\mathrm{d}[\sigma_t^2]}{\mathrm{d}t}} \, \mathrm{d}\bar{\mathbf{w}} \right\|_2 \tag{39}$$

$$\geq -\sigma_{\hat{t}}\mathbb{E}_{\epsilon_{\hat{t}}}\left[\|\epsilon_{\hat{t}}\|_2\right] + \left\| \int_{\hat{t}}^0 \left[ -\frac{\mathrm{d}[\sigma_t^2]}{\mathrm{d}t} \mathbf{s}_\theta(\boldsymbol{\tau}, t) \right] \mathrm{d}t \right\|_2 - \left\| \int_{\hat{t}}^0 \sqrt{\frac{\mathrm{d}[\sigma_t^2]}{\mathrm{d}t}} \, \mathrm{d}\bar{\mathbf{w}} \right\|_2 \tag{40}$$

$$\geq -\sqrt{d}\sigma_{\hat{t}} + (C + \Delta)\sigma_{\hat{t}}^2 - \left\| \int_{\hat{t}}^0 \sqrt{\frac{\mathrm{d}[\sigma_t^2]}{\mathrm{d}t}} \, \mathrm{d}\bar{\mathbf{w}} \right\|_2, \tag{41}$$

where we follow a similar procedure as in the proof for controlling type I error, except deriving the lower bound by employing the triangle inequality. Therefore, under alternative hypothesis, with probability at least $(1 - \delta)$, we have that:

$$\text{restoration gap}_{\hat{t},\theta}(\boldsymbol{\tau}) \geq b_{II} = \sigma_{\hat{t}} \left( (C + \Delta)\sigma_{\hat{t}} - \sqrt{d} - \sqrt{d + 2\sqrt{-d \cdot \log \delta} - 2\log \delta} \right) \tag{42}$$

which guarantee a bounded type II error at most $\delta$.

**Combining Type I and Type II Thresholds**

Following Theorem 3.3 in (Han et al., 2019), when $b_I \leq b_{II}$, selecting $b_I$ as the threshold ensures that both type I and type II errors are at most $2\delta$. Therefore, under

$$\Delta \geq \frac{2\sqrt{d} + 2\sqrt{d + 2\sqrt{-d \cdot \log \delta} - 2\log \delta}}{\sigma_{\hat{t}}} \tag{43}$$

setting

$$b \geq \sigma_{\hat{t}} \left( C\sigma_{\hat{t}} + \sqrt{d} + \sqrt{d + 2\sqrt{-d \cdot \log \delta} - 2\log \delta} \right) \tag{44}$$

guarantees both type I and type II errors at most $2\delta$, which completes the proof. $\qquad\square$

## Appendix B   Additional Experiments

### B.1   Sensitivity Analysis to Input Attribution Methods

We investigate how sensitive the proposed method is depending on the choice of input attribution methods. As described in Section 4.2, RGG+ utilizes the input attribution method as a regularizer to prevent adversarial restoration gap guidance. We apply the input attribution method to the gap predictor, where it quantifies the impact of each transition in the generated plan on the prediction of the restoration gap. To validate the robust-

Table 4: Performance of RGG+ in Maze2D depending on the choice of attribution maps which suggests the robustness of the attribution map regularizer.

|  | Maze2D Large | Multi2D Large |
|---|---|---|
| Grad-Cam | $143.9 \pm 1.50$ | $150.9 \pm 1.25$ |
| Saliency | $142.7 \pm 1.56$ | $150.1 \pm 1.29$ |
| DeepLIFT | $143.8 \pm 1.48$ | $150.9 \pm 1.26$ |

ness of RGG+ to different input attribution methods, we compare its performance while employing three attribution methods: Grad-CAM (Selvaraju et al., 2017), Sailency (Simonyan et al., 2013), and DeepLIFT (Shrikumar et al., 2017). The experimental results for Maze2D tasks are presented in Table 4. RGG+ exhibits comparable performances across the three attribution methods, which implies that our proposed method is robust in terms of the choice of input attribution methods.

## B.2 Sensitivity Analysis to Perturbation Magnitude

To further investigate the effect of choosing different $\hat{t}$ values on the restoration gap, we compare the performance of plans which are chosen up to top-K% from the restoration gap when setting $\hat{t}$ to 0.3, 0.5, and 0.7, in addition to the initial setting of 0.9. The results of this ablation study are illustrated in Figure 6. The results clearly show that the restoration gap remains significantly correlated with planning performance, even with these varied $\hat{t}$ values, when considering the top 10% of results. Specifically, while the correlation is strongest with $\hat{t}$ set to 0.9, even with $\hat{t}$ at 0.3, 0.5, or 0.7, the restoration gap still demonstrates a stronger correlation with planning performance compared to the rarity score

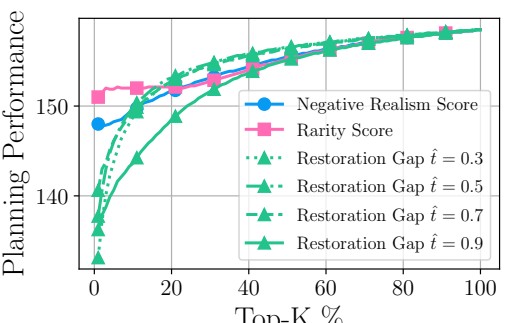

Figure 6: An ablation study varying $\hat{t}$ to understand the impact of the magnitude of the applied perturbation.

or the negative realism score. This indicates the robustness of the restoration gap as a metric for assessing the quality of generated plans. Moreover, these results support our Proposition 1 that a larger $\hat{t}$ is an effective choice when using the restoration gap.

## B.3 Comparison to Other Metrics

Table 5: Performance Comparison with rarity score and negative realism score.

| Environment | | Rarity | Negative Realism | RGG | RGG+ |
|---|---|---|---|---|---|
| Maze2D | Large | $126.9 \pm 2.1$ | $128.9 \pm 1.6$ | $\mathbf{135.4} \pm 1.7$ | $\mathbf{143.9} \pm 1.5$ |
| Multi2D | Large | $143.4 \pm 1.7$ | $143.3 \pm 1.5$ | $\mathbf{148.3} \pm 1.4$ | $\mathbf{150.9} \pm 1.3$ |

We investigate how effectively the restoration gap evaluates the quality of generated samples by comparing the histogram with various metrics in Figure 2 and comparing the planning performance in Figure 4. To further explore how significant a guidance signal provided by the restoration gap is for refining a diffusion planner, we compare the planning performance of plans guided by various metrics including the rarity score, the negative realism score, and the restoration gap. We conduct additional experiments in the Maze2D Large and Multi2D Large environments, the results of which are presented in Table 5. Consistent with the observations in the previous experiments, Table 5 clearly demonstrates that the restoration gap is a useful metric for control tasks. Unlike other metrics requiring expert data for training, our restoration gap works without such constraints, making it even more practical.

## B.4 Comparison to Discriminator Guidance

In our work, we have demonstrated the effectiveness of utilizing the restoration gap prediction model to enhance the performance of Diffuser. However, there could be an alternative approach that is

Table 6: Performance Comparison with discriminator guidance (DG).

| Environment | | Diffuser | DG | RGG | RGG+ |
|---|---|---|---|---|---|
| Maze2D | Large | $123.5 \pm 2.0$ | $127.0 \pm 1.9$ | $\mathbf{135.4} \pm 1.7$ | $\mathbf{143.9} \pm 1.5$ |
| Multi2D | Large | $141.2 \pm 1.6$ | $143.6 \pm 1.6$ | $\mathbf{148.3} \pm 1.4$ | $\mathbf{150.9} \pm 1.3$ |

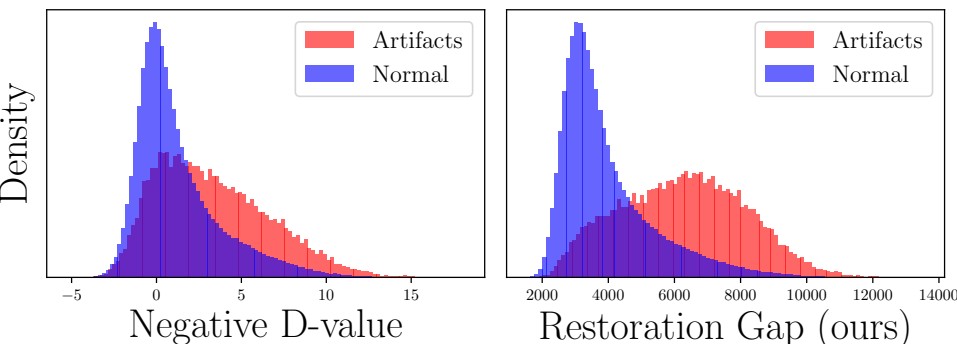

Figure 7: The distribution differences between Artifacts and Normal plans illustrated through the density of the negative D-value and restoration gap.

to leverage the output of the discriminator which distinguishes whether the given plan is real or generated; we refer to this discriminator output as the "D-value". It is plausible as the discriminator inherently distinguishes between real and generated trajectories.

To evaluate the ability of the discriminator to accurately identify infeasible plans, we define a subset of plans generated by Diffuser (Janner et al., 2022) as "artifact plans". These artifact plans consist of transitions that include passing through walls, an impossible action for the agent to follow. We then compare the distribution of the restoration gap for both the normal and artifact plans.

As illustrated in Figure 7, our comparison reveals that the D-value fails to distinguish between artifacts and normal groups as effectively as the restoration gap. A noteworthy observation is that the discriminator, while adept at identifying infeasible transitions within generated trajectories, tends to focus more on local transitions rather than the overall structure. This local concentration results in it being less useful than the restoration gap when it comes to recognizing infeasible plans.

To further illustrate the practical implications of these findings, we conduct an additional performance comparison experiment. Here, the discriminator guidance (DG) is used to refine the Diffuser, and the resulting performances are compared with those of RGG and RGG+ (see Table 6). In our comparative experiments involving Maze2D Large and Multi2D Large environments, it is apparent that the restoration gap guidance methods significantly outperform both the original Diffuser and the discriminator guidance (DG) method. While DG offers a slight improvement over Diffuser, it is unable to match the performance enhancements provided by our proposed methods.

These results underscore the effectiveness of the restoration gap as a reliable metric for improving trajectory generation by diffusion planners. This superiority holds even when compared to discriminator guidance, which directly capitalizes on the discriminator's capability to distinguish between real and generated trajectories. Consequently, it is evident that the restoration gap provides a more efficient strategy for refining diffusion planners.

### B.5 Visualization of Plans with Low and High Restoration Gap

To understand how restoration gap values relate to the qualities of generated plans and to validate the efficacy of the restoration gap in detecting infeasible plans, we present additional qualitative results comparing visualized plans with low and high restoration gaps.

In the Maze2D-Large environment, as depicted in Figure 8, plans with low restoration gap values exhibit smoother and more coherent trajectories. In contrast, plans with high restoration gap values often include physically infeasible transitions, such as passing through walls or abrupt changes in direction.

In the HalfCheetah environment, as demonstrated in Figure 9, the instability of movement, particularly during landing, is more apparent in plans with a high restoration gap. On the other hand, plans with low restoration gap values allow for stable landings, enabling the cheetah to move farther.

In the Unconditional Block Stacking environment, as illustrated in Figure 10, plans with a high restoration gap exhibit error-prone transitions. For example, in such plans, the robotic arm suddenly teleports from the initial joint position, magically grasps a block stacked under other blocks, or unexpectedly changes the grasped block. Moreover, these plans violate physical constraints, such as having the block in the same position as the robotic arm. In comparison, plans with low restoration gap values comply with the physical constraints.

This disparity between plans with low and high restoration gap values illustrates the effectiveness of our restoration gap metric in distinguishing between reliable and unreliable plans across various environments.

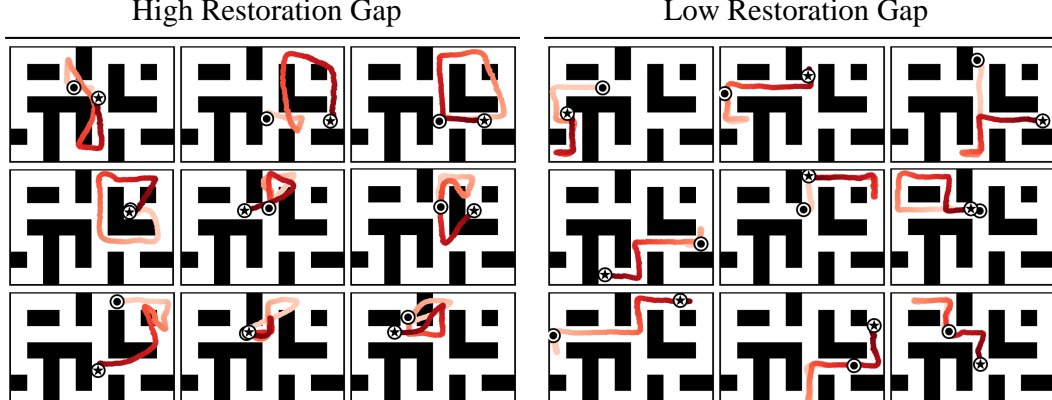

Figure 8: Visual comparison of plans of low and high restoration gap values generated by Diffuser in the Maze2D-Large environment.

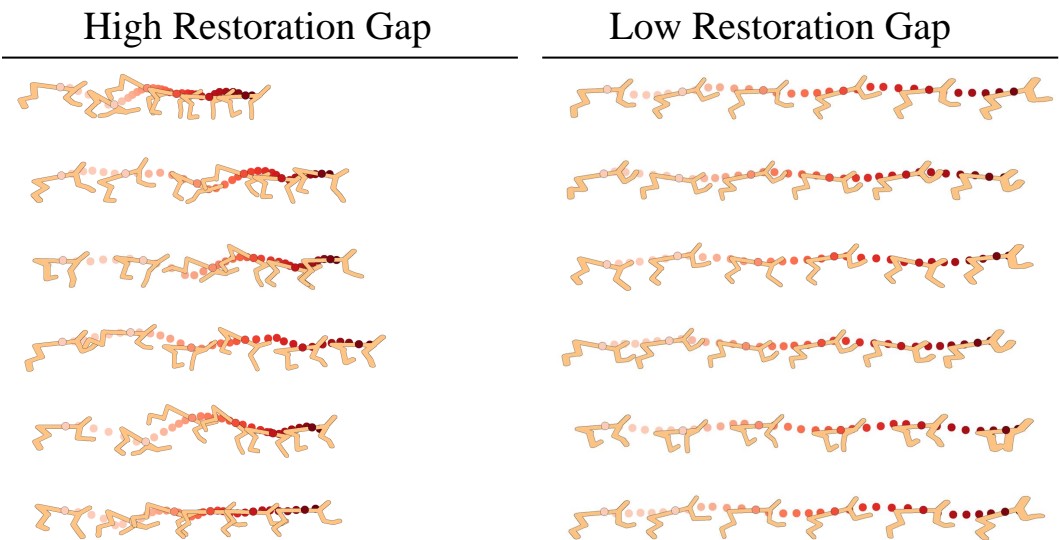

Figure 9: Visual comparison of plans of low and high restoration gap values generated by Diffuser in the HalfCheetah environment.

| High Restoration Gap | Low Restoration Gap |
|:---:|:---:|
| planning horizon → | planning horizon → |

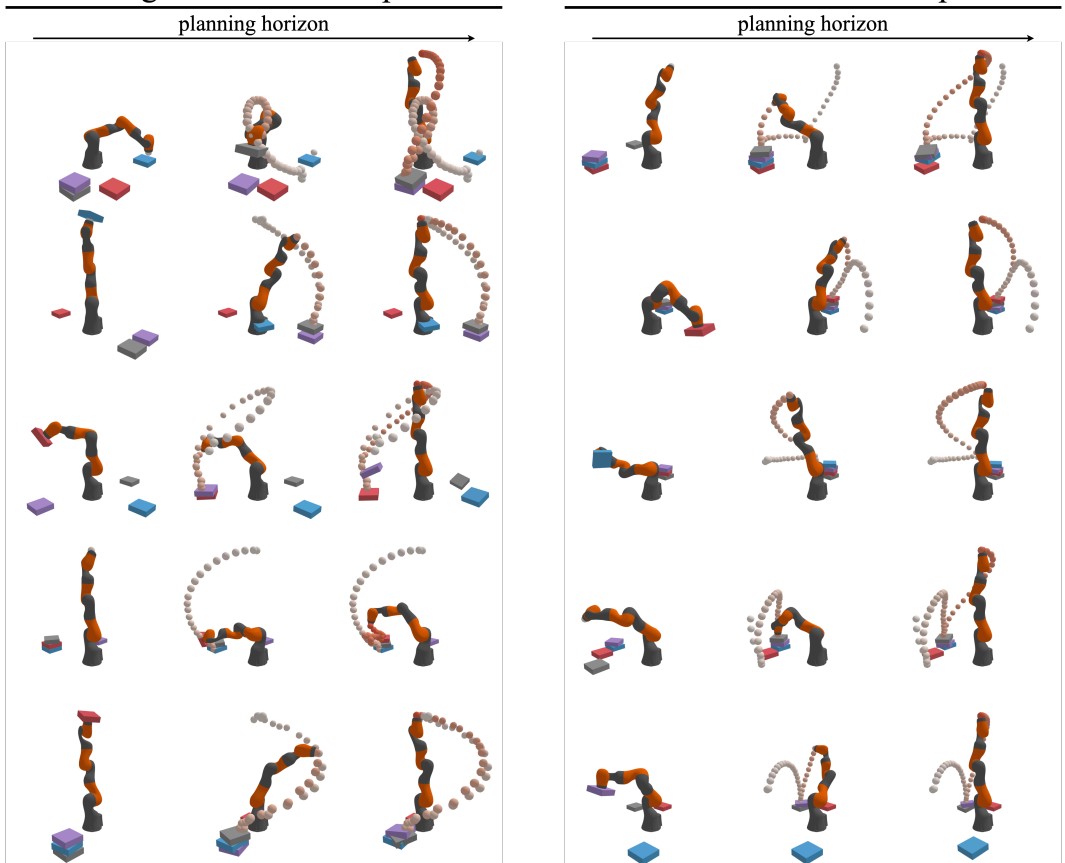

Figure 10: Visual comparison of plans of low and high restoration gap values generated by Diffuser in the Unconditional Block Stacking environment.

## Appendix C    Experimental Setup Details

### C.1    Environments

**Maze2D**    Maze2D environments (Fu et al., 2020) involve a navigation task where an agent needs to plan for a long-horizon to navigate toward a distant target goal location. A reward is not provided except when the agent successfully reaches the target goal where it only gets a reward of 1. Maze2D environments consist of three distinct maze layouts: "U-Maze", "Medium", and "Large" each of which offers different levels of difficulty. In Maze2D environments, there are two tasks: a single-task, where the goal location is fixed, and a multi-task, which we refer to as Multi2D, where the goal location is randomly selected at the beginning of every episode. Details about Maze2D environments are summarized in Table 7.

Table 7: Environment details for Maze2D experiments.

|  | **Maze2D-Large** | **Maze2D-Medium** | **Maze2D-UMaze** |
|:---:|:---:|:---:|:---:|
| State space $\mathcal{S}$ | $\in \mathbb{R}^4$ | $\in \mathbb{R}^4$ | $\in \mathbb{R}^4$ |
| Action space $\mathcal{A}$ | $\in \mathbb{R}^2$ | $\in \mathbb{R}^2$ | $\in \mathbb{R}^2$ |
| Goal space $\mathcal{G}$ | $\in \mathbb{R}^2$ | $\in \mathbb{R}^2$ | $\in \mathbb{R}^2$ |
| Episode length | 800 | 600 | 300 |

**Locomotion**    Gym-MuJoCo locomotion tasks (Fu et al., 2020) are widely used benchmarks for evaluating algorithms on heterogeneous data with varying quality. The "Medium" dataset is generated

by collecting 1M samples from an SAC agent (Haarnoja et al., 2018) trained to approximately one-third of the performance level compared to an expert. The "Medium-Replay" dataset includes all samples acquired during training until a "Medium" level of performance is achieved. The "Medium-Expert" dataset is composed of an equal mixture of expert demonstrations and sub-optimal data. Environmental details for Locomotion experiments are summarized in Table 8.

Table 8: Environment details for Locomotion experiments.

|  | **Hopper-*** | **Walker2d-*** | **Halfcheetah-*** |
|---|---|---|---|
| State space $\mathcal{S}$ | $\in \mathbb{R}^{11}$ | $\in \mathbb{R}^{17}$ | $\in \mathbb{R}^{17}$ |
| Action space $\mathcal{A}$ | $\in \mathbb{R}^{3}$ | $\in \mathbb{R}^{6}$ | $\in \mathbb{R}^{6}$ |
| Episode length | 1000 | 1000 | 1000 |

**Block Stacking**  The Block Stacking suite is a benchmark used to evaluate the model performance for a large state space, which employs a Kuka iiwa robotic arm (Janner et al., 2022). The offline demonstration data required to train a policy model or a diffusion planning model is obtained through the application of PDDLStream (Garrett et al., 2020).

The objective of the unconditional stacking task is to construct a tower of blocks with the maximum possible height. In this task, the agent observes the state including the joint position of the robot, as well as the position and rotation of each block, and then commands the robot's desired joint position while performing the grasping action to pick up the blocks. In the conditional stacking task, where the objective is to stack blocks in a specified order, the agent observes the same state as in the unconditional stacking task, but it additionally observes the index of the block which indicates the order in which the blocks should be stacked.

We employ the same diffusion model for both tasks, but in the conditional stacking task, we additionally utilize a value function to guide the diffusion planner in stacking blocks according to specified conditions. Details about Block Stacking environments are summarized in Table 9.

Table 9: Environment details for Block Stacking experiments.

|  | **Unconditional Stacking** | **Conditional Stacking** |
|---|---|---|
| State space $\mathcal{S}$ | $\in \mathbb{R}^{39}$ | $\in \mathbb{R}^{43}$ |
| Action space $\mathcal{A}$ | $\in \mathbb{R}^{11}$ | $\in \mathbb{R}^{11}$ |
| Episode length | 384 | 384 |

**C.2  Other Metrics for the Assessment of Individually Generated Samples**

Kynkäänniemi et al. (2019) present the notions of improved precision and recall to the study of generative models. Precision is determined by examining if each generated sample falls within the estimated manifold of real samples. Symmetrically, recall is computed by checking if each real sample resides within the estimated manifold of generated samples. Real and generated sample feature vectors are represented as $\phi_r$ and $\phi_g$, respectively, and the corresponding sets of these feature vectors are denoted by $\mathbf{\Phi}_r$ and $\mathbf{\Phi}_g$. To compute improved precision and recall, the manifolds of real and generated samples are estimated using the sets of $k$-NN hyperspheres for each sample:

$$\text{manifold}_k(\mathbf{\Phi}) = \bigcup_{\phi' \in \mathbf{\Phi}} B_k(\phi', \mathbf{\Phi}), \quad B_k(\phi', \mathbf{\Phi}) = \{\phi \mid \|\phi' - \phi\|_2 \leq \|\phi' - \text{NN}_k(\phi', \mathbf{\Phi})\|_2\} \quad (45)$$

where $\text{NN}_k(\phi', \mathbf{\Phi})$ returns the $k$th nearest feature vector of $\phi'$ from the set $\mathbf{\Phi}$. $B_k(\phi', \mathbf{\Phi})$ is the $k$-NN hyperspheres with the radius of $\|\phi' - \text{NN}_k(\phi', \mathbf{\Phi})\|_2$.

Although the improved precision metric provides a way to evaluate the quality of a population of generated samples, it yields only a binary result for an individual sample, making it unsuitable for ranking individual samples by their quality. In contrast, realism score (Kynkäänniemi et al., 2019) and rarity score (Han et al., 2023) offer a continuous extension of improved precision and recall, enabling the assessment of individually generated sample quality.

**Realism Score** The realism score quantifies the maximum inverse relative distance of a generated sample within a $k$-NN hypersphere originating from real data.

$$\text{realism score}(\phi_g, \mathbf{\Phi}_r) = \max_{\phi_r} \frac{\|\phi_r - \text{NN}_k(\phi_r, \mathbf{\Phi}_r)\|_2}{\|\phi_g - \phi_r\|_2}. \tag{46}$$

A high realism score is achieved when the relative distance between a generated sample and a real sample is small, compared to the radius of the real sample's $k$-NN hypersphere.

**Rarity Score** The rarity score measures the radius of the smallest nearest-neighbor sphere that contains the generated sample.

$$\text{rarity score}(\phi_g, \mathbf{\Phi}_r) = \min_{r, s.t. \phi_g \in B_k(\phi_r, \mathbf{\Phi}_r)} \|\phi_r - \text{NN}_k(\phi_r, \mathbf{\Phi}_r)\|_2. \tag{47}$$

This is grounded in the hypothesis that normal samples will be closely grouped, whereas unique and rare samples will be dispersed in the feature space.

In an attempt to compare the restoration gap against negative realism scores and rarity scores, we apply the parameters suggested in (Han et al., 2023), making use of $k = 3$ and 30,000 real samples to approximate the real manifold and calculate the corresponding scores: negative realism and rarity.

## Appendix D    Implementation Details and Hyperparameters

### D.1    Implementation of Gap Predictor

We employ a temporal U-Net architecture, with repeated convolutional residual blocks, for parameterizing $\mathcal{G}_\psi$ as introduced in Diffuser (Janner et al., 2022). By using the pre-trained down blocks from Diffuser's diffusion model as our feature extraction module and keeping it fixed during training, we can achieve enhanced performance and reduce training costs. The hyperparameters for training the gap predictor are summarized in Table 10.

Table 10: Hyperparameters used for training the gap predictor. Values that are within brackets are separately tuned through a grid search.

| Hyperparameter | Maze2D | Locomotion | Block Stacking |
|:---:|:---:|:---:|:---:|
| # synthetic data | 500000 | 500000 | 500000 |
| Observation normalization | Yes | Yes | Yes |
| Gap predictor learning rate | 0.0002 | 0.0002 | {0.00002, **0.0002**, 0.001} |
| Gap predictor batch size | 32 | 32 | {32, 64, **128**, 256} |
| Gap predictor train steps | 2000000 | 2000000 | 2000000 |
| $\hat{t}$ for perturbation magnitude | 0.9 | 0.9 | 0.9 |

### D.2    Implementation of Restoration Gap Guidance

In this section, we document hyperparameters employed for restoration gap guidance. We adopt the hyperparameters from Diffuser (Janner et al., 2022) for determining the planning horizon and diffusion steps. Furthermore, the values of $\alpha$, $\beta$, and $\lambda$ are determined through a grid search. The hyperparameters utilized for the Maze2D experiments are presented in Table 11, while those for the Locomotion experiments are summarized in Table 12, 13, and 14. The hyperparameters for the Block Stacking experiments are provided in Table 15.

The planning performance exhibits a clear trend based on the choices of $\alpha$, $\beta$, and $\lambda$ parameters. This allows us to perform a grid search using the relatively minimal number of evaluation episodes. As depicted in Figure 3, the planning performance initially increases with rising values of $\lambda$ but begins to decline when the values become excessively large. Specifically, we conduct 10, 15, and 10 evaluation episodes for the Maze2D, Locomotion, and Block Stacking experiments, respectively.

Table 11: Specific hyperparameters for Maze2D experiments. Values that are within brackets are separately tuned through a grid search.

| Hyperparameter | Large | Medium | UMaze |
|---|---|---|---|
| Planning horizon | 384 | 256 | 128 |
| Diffusion steps | 256 | 256 | 64 |
| # samples for MC estimate of restoration gap | 10 | 10 | 10 |
| $\alpha$ for scaling the overall guidance | {**0.05**, 0.1} | {**0.05**, 0.1} | {**0.05**, 0.1} |
| $\beta$ for scaling restoration gap guidance | 1.0 | 1.0 | 1.0 |
| $\lambda$ for scaling attribution map regularization | {0.1, 1.0, **3.0**} | {**0.1**, 1.0, 3.0} | {0.1, 1.0, **3.0**} |

Table 12: Specific hyperparameters for HalfCheetah experiments. Values that are within brackets are separately tuned through a grid search.

| Hyperparameter | Med-Expert | Medium | Med-Replay |
|---|---|---|---|
| Planning horizon | 4 | 32 | 32 |
| Diffusion steps | 20 | 20 | 20 |
| # samples for MC estimate of restoration gap | 64 | 64 | 64 |
| $\alpha$ for scaling the overall guidance | {**0.01**, 0.1} | {0.01, **0.1**} | {**0.01**, 0.1} |
| $\beta$ for scaling restoration gap guidance | {0.1, 1.0, **10.0**} | {0.1, **1.0**, 10.0} | {0.1, 1.0, **10.0**} |
| $\lambda$ for scaling attribution map regularization | {0.001, 0.01, 0.1, **1.0**} | {0.001, 0.01, 0.1, **1.0**} | {0.001, **0.01**, 0.1, 1.0} |

Table 13: Specific hyperparameters for Hopper experiments. Values that are within brackets are separately tuned through a grid search.

| Hyperparameter | Med-Expert | Medium | Med-Replay |
|---|---|---|---|
| Planning horizon | 32 | 32 | 32 |
| Diffusion steps | 20 | 20 | 20 |
| # samples for MC estimate of restoration gap | 64 | 64 | 64 |
| $\alpha$ for scaling the overall guidance | {0.01, **0.1**} | {0.01, **0.1**} | {0.01, **0.1**} |
| $\beta$ for scaling restoration gap guidance | {0.1, **1.0**, 10.0} | {0.1, **1.0**, 10.0} | {0.1, 1.0, **10.0**} |
| $\lambda$ for scaling attribution map regularization | {**0.001**, 0.01, 0.1, 1.0} | {0.001, 0.01, **0.1**, 1.0} | {0.001, 0.01, 0.1, **1.0**} |

Table 14: Specific hyperparameters for Walker2d experiments. Values that are within brackets are separately tuned through a grid search.

| Hyperparameter | Med-Expert | Medium | Med-Replay |
|---|---|---|---|
| Planning horizon | 32 | 32 | 32 |
| Diffusion steps | 20 | 20 | 20 |
| # samples for MC estimate of restoration gap | 64 | 64 | 64 |
| $\alpha$ for scaling the overall guidance | {**0.01**, 0.1} | {**0.01**, 0.1} | {0.01, **0.1**} |
| $\beta$ for scaling restoration gap guidance | {0.1, **1.0**, 10.0} | {0.1, 1.0, **10.0**} | {**0.1**, 1.0, 10.0} |
| $\lambda$ for scaling attribution map regularization | {0.001, 0.01, 0.1, **1.0**} | {0.001, 0.01, 0.1, **1.0**} | {**0.001**, 0.01, 0.1, 1.0} |

Table 15: Specific hyperparameters for Block Stacking experiments. Values that are within brackets are separately tuned through a grid search.

| Hyperparameter | Unconditional Stacking | Conditional Stacking |
|---|---|---|
| Planning horizon | 128 | 128 |
| Diffusion steps | 1000 | 1000 |
| # samples for MC estimate of restoration gap | 5 | 5 |
| $\alpha$ for guide scale | {0.01, **0.02**, 0.05, 0.1, 0.2, 0.5} | {0.1, 0.3, **0.5**, 0.7} |
| $\beta$ for scaling restoration gap guidance | 1.0 | {**0.01**, 0.02, 0.04, 0.07} |
| $\lambda$ for scaling attribution map regularization | {0.1, **0.3**, 0.5, 1.0, 3.0, 5.0} | {0.001, **0.003**, 0.005, 0.008} |

### D.3 Implementation of Attribution Map Regularizer

For the Maze2D experiments and the Block Stacking experiments, we adopt Grad-CAM (Selvaraju et al., 2017), while we employ DeepLIFT (Shrikumar et al., 2017) for the Locomotion experiments, as the attribution method. The rationale for the choice of each method is that both Grad-CAM and DeepLIFT offer simplicity and efficiency in computation. Specifically, we apply the DeepLIFT for

the Locomotion experiments due to their relatively smaller planning horizon compared to the other tasks. This is because Grad-CAM compresses saliency information into a single value. However, it is worth noting that our proposed method has shown robustness across different input attribution methods, as demonstrated in Table 4.

### D.4 The Amount of Computation

Our proposed method, refining diffusion planner, requires training the gap predictor which estimates the restoration gap. For this purpose, we generate synthetic data of 500,000 plans generated by Diffuser. The generation process takes approximately 30 to 50 hours, depending on the situation, on a single NVIDIA Quadro 8000 GPU. The training time of the gap predictor on the same GPU can range from 5 to 8 hours.

## Appendix E   Baseline Performance Sources

### E.1 Maze2D Tasks

The scores for CQL are taken from Table 2 in Fu et al. (2020). The scores for MPPI and IQL are taken from Table 1 in Janner et al. (2022).

### E.2 Locomotion Tasks

The scores for BC, CQL, and IQL are found in Table 1 of Kostrikov et al. (2022), while DT scores are taken from Table 2 in Chen et al. (2021), TT from Table 1 in Janner et al. (2021), MOPO from Table 1 in Yu et al. (2020), MOReL from Table 2 in Kidambi et al. (2020), MBOP from Table 1 in Argenson & Dulac-Arnold (2021), and Diffuser from Table 2 in Janner et al. (2022).

### E.3 Block Stacking Tasks

The scores corresponding to BCQ and CQL are obtained from Table 3 of Janner et al. (2022). To obtain the scores for Diffuser, the official implementation and model provided by the authors are used, which can be found at `https://github.com/jannerm/diffuser`.

