# OpenReview forum: "Refining Diffusion Planner for Reliable Behavior Synthesis by Automatic Detection of Infeasible Plans"
_NeurIPS.cc/2023/Conference — NeurIPS 2023 poster_

### Official Review · Reviewer_9DAN · 2023-07-03

**Soundness:** 2 fair
**Presentation:** 3 good
**Contribution:** 1 poor
**Rating:** 5
**Confidence:** 4

**Summary:**

In this work, authors propose a new metric called “restoration gap” that is able to measure the reliability of plans emerging from diffusion-based planners. They then suggest using this metric along with guidance, “restoration guidance”, to encourage the diffusion-based planner to generate more reliable plans. In addition, they attempt to improve the performance of restoration guidance by using an attribution map regularizer. Finally, they evaluate their method on three different offline control benchmarks.

**Strengths:**

Overall Strengths:
1. The problem of detecting and/or improving unreliable plans generated by diffusion-based planners is both interesting and important and I appreciate the authors for their efforts towards this direction.
2. Paper is well-written and well-presented with good quality visualizations, plots and tables.
3. Experiments are covering a good-enough domain of tasks and compare against a large number of benchmarks.


**Weaknesses:**

Overall Weaknesses:
1. A critical weakness, in my view, is the lack of substantial improvement demonstrated by the proposed solution compared to the Diffuser benchmark. Across various experiments, RGG exhibits only marginal enhancements (if any) over Diffuser, and similarly, RGG+ shows only minor improvements (if any) over RGG. This observation raises two possibilities: either the solution itself lacks significant impact or the current experimental setup fails to adequately capture the method's importance. While I appreciate the authors' overall research trajectory, it is my belief that this particular submission requires further development and refinement.
2. Additionally, there are some additional limitations in the work. Firstly, to establish the significance of their approach, it would be valuable to investigate how it compares to rejection-based methods, especially in scenarios where the problem context remains fixed. For instance, in the context of the Maze2D experiments, the primary objective is to find solutions that navigate around obstacles. One potential approach could involve incorporating collision-detection code and subsequently discarding samples that do not satisfy this condition.
3. Furthermore, to demonstrate the significance of the proposed metric, it would be valuable to show how guidance with restoration gap compares with guidance with realism score and rarity score (or other readily avaible metrics).



**Questions:**

Overall, the paper is well-written and easy to follow. However, there are some parts that need improvement. For me, these include:
- In equation 4, the addition of trajectory optimization is vague. Maybe, a reference to section 3.2 of Janner et al. 2022 where trajectory optimality in diffusion models is introduces could be beneficial.
- In equation 5, g is defined but is not explained what it is supposed to be and how it is supposed to be used. For someone new to the field, this is problematic.
- In figure 2, the realism gap and rarity score are shown alongside restoration gap but they are not explained until section 5.1. I recommend moving this explanation to the caption of figure 2 or to section 3.
- In section 3, it is explained why restoration gap is an intuitive measure of plan reliability. It would be better if you could clarify what reliability really means. Are reliable plans ones that are within distribution? Moreover, a discussion on how much noise is required is also essential as too much noise may push the recovered samples too far from the original and too little will not provide any improvements.
- Attribution map is a good option for showing the explainability of diffusion-planner with the trained gap predictor. However, its significance as a regularizer is not supported by the experimental results. Therefore, I would remove section 4.2 as it doesn’t provided any added value.
- Overall, the enhancements offered by the new method (RGG and RGG+) in comparison to Diffuser do not appear to be substantial, if any. As a result, the overall significance of the work is diminished in my perspective.
- Following section 5.1, it would be valuable to show how guidance with restoration gap compares with guidance with realism score and rarity score. This could potentially support the fact that the newly proposed metric is more suitable for the task than other available metrics.
- Throughtout the paper, it is claimed that the proposed approach for improving unreliable plans emerging from diffusion-based planners is novel but this claim is not supported. Firstly, the provided solution which is to use guidance based on some form of reliability-measuring metric, is not necessarily novel and it seems to be the immediate and instinctive response to addressing the issue. Moreover, the newly proposed metric which is based on the addition of noise for sample recovery and purification has been explored before as well (see SDEdit by Meng et al. and Diffusion Models for Adversarial Purification by Nie et al.). Therefore, the authors’ claim in the approach being novel has to either be removed or clarified.


**Limitations:**

In addition to the recommendations on the specific technical parts (7 bullet points in “Questions”), I highly recommend developing the idea of purifying diffuser-generated plans further using various techniques and comparing what provides the best performance. Moreover, designing an experimental setup that captures the significance of reliability of plans is highly beneficial to support the significance and impact of the work.

---

> ### Author Rebuttal · Authors · 2023-08-09
>
> **Q: Comparison to rejection-based methods, especially in scenarios where the problem context remains fixed.**
>
> **A:** We respectfully disagree with the reviewer’s comment on comparing our approach to rejection-based methods. It is important to note that rejection-based methods rely on strong assumptions, such as requiring the oracle understanding of the environment, which is uncommon in general machine learning scenarios.
>
> For example in the context of Maze2D environments, incorporating collision-detection code would necessitate the agent to have an understanding of the location of every wall in the environment where it interacts. However, in Maze2D environments, the agent only observes its 2D position, which lacks information about the location of walls.
>
> We appreciate the reviewer’s suggestion, but considering the strong assumptions and the context of our approach, comparing to rejection-based methods may not be a suitable avenue of investigation.
>
> **Q: In equation 4, the addition of trajectory optimization is vague.**
>
> **A:** Thank you for pointing out the ambiguity in Equation 4. We’ll add a reference to Section 3.2 of Janner et al. (2022) for clarity on trajectory optimization in our revised version.
>
> **Q: In equation 5, g is defined but is not explained.**
>
> A: $g$ is defined to be the derivative of objective value $\mathcal{J}$ as represented in Equation 5. However, as there is no explicit description on this, we’ll include it in the revised version.
>
> **Q: In figure 2, the realism gap and rarity score are shown alongside restoration gap but they are not explained until section 5.1.**
>
> **A:** Due to page constraints, we couldn’t initially include the description of them in Section 3. In the camera-ready version that allows an additional page, we’ll ensure the explanations align better with Figure 2 for clarity.
>
> **Q: Are reliable plans ones that are within distribution? Moreover, a discussion on how much noise is required is also essential.**
>
> **A:** In the context of our work, a plan’s **reliability** indicates its feasibility in execution and its alignment with the training distribution. Therefore, yes, reliable plans are ones that are within the distribution, assuming a sufficient amount of offline dataset.
>
> According to Proposition 1, minimizing both type I error (false positives) and type II error (false negatives) requires a substantial $\sigma_{\hat{t}}$, which, in turn, a large $\hat{t}$. We empirically observe that the $\hat{t}=0.9$ works well for all tasks.
>
> Furthermore, in our ablation study, we explored how sensitive the restoration gap is to the value of $\hat{t}$ as shown in Figure 6 in Appendix C.2. We compared the performance of plans that are chosen from the restoration gap when setting $\hat{t}$ to 0.3, 0.5, 0.7, and 0.9, in addition from the negative realism score and rarity score. As a result, we observed that the restoration gap is robust to $\hat{t}$ though $\hat{t}=0.9$ performs the best which supports Proposition 1. Moreover, the restoration gap demonstrated superior correlation with the planning performance over other metrics.
>
> **Q: Significance of attribution map as a regularizer is not supported by the experimental results.**
>
> **A:** We respectfully disagree with your assessment.
>
> The beneficial effect of RGG+ as a regularizer is clearly illustrated in Figure 3 in Section 4.2, which shows the performance of RGG+ in the Maze2D-Large task across a range of $\lambda$ values. In the absence of regularization ($\lambda=0$), our diffusion planner is prone to generate adversarial plans which lead to a decline in performance as the planning budget increased. On the other hand, RGG+ maintains its effectiveness over a broad range of $\lambda$ values, where any $\lambda>0$ consistently outperforms $\lambda=0$, indicating the advantage of regularization. Moreover, RGG+ performs better than RGG across a wide spectrum of tasks including Maze2D, Locomotion, and Block Stacking. Hence, we believe that the content of Section 4.2 is non-trivial to our work and should not be omitted.
>
> **Q: The authors’ claim in the approach being novel has to either be removed or clarified.**
>
> **A:** We respectfully disagree. Adding a small amount of noise and then performing sample recovery through a reverse generative process has been explored in tasks like image synthesis (Meng et al., 2021) and adversarial purification (Nie et al., 2022). However, these works do not provide a way for evaluating the quality of individually generated samples produced by a diffusion model. Our work uniquely proposes a metric for assessing trajectory quality created by diffusion models, along with a novel refining guidance approach to address the limitations of diffusion-based planning.
>
> Furthermore, other reviewers appear to align with our perspective on novelty. Reviewer iw4y noted that our work appears to be novel, reviewer N3Wq recognized our novel approach as contributing to the field and expanding understanding of plan refinement, and reviewer pN8B stated that our contribution to the paper is solid, recognizing the restoration gap as an intuitive metric for measuring trajectory quality.
>
> We firmly believe that the novelty of our approach, which focuses on evaluating and refining trajectory quality generated by diffusion models, is non-trivial and has the potential to contribute meaningfully to the field.
>
> **Q: Lack of substantial improvement demonstrated by the proposed solution.**
>
> **A:** Please refer to the global response provided above.
>
> **Q: Show how guidance with restoration gap compares with guidance with realism score and rarity score.**
>
> **A:** Please refer to the global response provided above.
>
> **References**
>
> - Janner et al., Planning with Diffusion for Flexible Behavior Synthesis. In *ICML*, 2022.
> - Meng et al., Sdedit: Guided image synthesis and editing with stochastic differential equations. In *ICLR*, 2021.
> - Nie et al., Diffusion models for adversarial purification. In *ICML*, 2022.

---

> > ### Comment · Reviewer_9DAN · 2023-08-13
> >
> > I appreciate the authors' thorough response during the rebuttal process and applying the suggested changes.
> >
> > I agree that the experimental results with the additional context are now more noteworthy.
> >
> > Regarding novelty, I appreciate the discussion provided by the authors. While I understand that the application of an existing technique to a new domain may yield novel insights, I must clarify that *in my view*, novelty also encompasses the introduction of genuinely new concepts that significantly contribute to the advancement of a particular problem domain. I would suggest to tone down the claims of novelty in the paper to ensure a clear and accurate representation of the work's contributions. I acknowledge that the notion of novelty can vary between reviewers, and I respect that other reviewers have expressed their recognition of the novel aspects of the approach. However, I still feel it is important to emphasize that my assessment of novelty is based on the understanding that introducing new techniques or concepts is a fundamental aspect of what constitutes novelty.
> >
> > I still have some confusion regarding the main motivation of the paper which is to target infeasible plans using restoration gap. The diffusion model will failt to reconstruct plans that are out of distribution, however infeasible plans and OOD plans are not necessarily the same. Therefore, I agree with other reviewers and I strongly suggest distinguishing between being feasible and being OOD in the paper and why the current set-up targets both.
> >
> > After thoughtful contemplation of the feedback provided by both the authors and my fellow reviewers, I am leaning towards adjusting my initial score for the paper. While I maintain the belief that the concerns raised in my original review are valid, the thorough discussion regarding infeasible versus OOD plans, along with the additional context provided for the experimental results, has led me to view this work more positively.

---

> > > ### Author Response · Authors · 2023-08-20
> > >
> > > We are pleased to hear that you acknowledge that our experimental results are more significant with the added context.
> > >
> > > Regarding the aspect of novelty, we appreciate your perspective, and we understand the distinction you've made between applying existing techniques to new domains and introducing entirely novel concepts. While we agree that our approach is not a genuinely new concept, we believe that the contributions of the proposed metric with theoretical justification and consequent application on refining diffusion planners are not subtle and hold substantial merit. To address your viewpoint, we will elaborate on related concepts (Meng et al., 2021; Nie et al., 2022) and highlight the distinguishing features of our approach in the Related Work section.
> > >
> > > In regard to the main motivation of our work, we would like to clarify what infeasible plans are once again and elaborate on how the restoration gap identifies those plans. We’ll also discuss how the restoration gap is related to OOD-ness.
> > >
> > > Infeasible plans are those that are incompatible with system constraints and cannot be executed. The high-level motivation behind identifying such plans is that the diffusion model would fail to reconstruct OOD plans. It motivates the restoration gap to perturb the initial plan and then reconstruct it.
> > >
> > > As we discussed with Reviewer iw4y, the restoration gap implicitly addresses infeasible plans through the lens of OOD-ness. By utilizing the diffusion planner’s temporal compositionality property, which enables the stitching of in-distribution subsequences, we argue that most infeasible plans are likely to be OOD, while feasible plans would generally remain within the distribution. As such, the restoration gap becomes a tool for identifying infeasible plans through their OOD characteristics. In response to your feedback, we will make sure to elaborate on the relationship between the restoration gap and OOD-ness.
> > >
> > > We genuinely appreciate the time and effort you've dedicated to thoroughly review our manuscript and offer your valuable insights. We sincerely hope that this response, coupled with our earlier rebuttal, has effectively addressed your concerns.
> > >
> > > If you have any further suggestions or concerns, please feel free to share them with us before the discussion period concludes.
> > >
> > > Once again, we extend our gratitude for your thoughtful consideration and feedback.
> > >
> > > **References**
> > >
> > > - Meng et al., Sdedit: Guided image synthesis and editing with stochastic differential equations. In *ICLR*, 2021.
> > > - Nie et al., Diffusion models for adversarial purification. In *ICML*, 2022.

---

### Official Review · Reviewer_mq5w · 2023-07-05

**Soundness:** 3 good
**Presentation:** 4 excellent
**Contribution:** 3 good
**Rating:** 7
**Confidence:** 3

**Summary:**

This paper investigates the potential of diffusion models in addressing long-horizon sparse reward tasks. While diffusion models are effective generative models, they are not inherently guaranteed to generate feasible plans. To overcome this limitation, the study proposes an approach to refine unreliable paths generated by diffusion models and provides guidance to enhance error-prone plans.
The evaluation of individual plans generated by diffusion models is done using a novel metric called the "restoration gap" which measures the restorability of a given path. The "restoration gap" metric effectively identifies plans with a low error probability, which is considered an indicator of plan feasibility. This metric is estimated using a function approximator called the "gap predictor."
With the help of the gap predictor, a Restoration Gap Guidance (RGG) is defined, which refines diffusion planners and improves the low-quality plans generated by diffusion model.
Moreover, the gap predictor can lead the agent in an undesirable direction due to estimation errors during the denoising process. Therefore, RGG is augmented with a regularizer to prevent plans from heading in the wrong direction, and this modified RGG is referred to as RGG+.
The proposed approach is evaluated on three different benchmarks. The experimental results demonstrate that the approach enhances planning performance compared to the benchmarks.

Overall, the paper presents a novel method for refining unreliable plans generated by diffusion models, improving the planning performance in long-horizon tasks. The approach is demonstrated to be effective in various offline control settings and provides explainability through the use of attribution maps.


**Strengths:**

Novel Approach: The paper proposes a novel approach to refining unreliable plans generated by diffusion models. It introduces the concept of restoration gap as a metric to evaluate the quality of individual plans generated by the diffusion model.

Theoretical Justification: The restoration gap metric is theoretically justified, and the paper provides an analysis of its properties. It demonstrates that the restoration gap can effectively detect artifacts (infeasible plans) with bounded error probabilities.

Improvement in Planning Performance: The proposed approach, called Restoration Gap Guidance (RGG), is shown to enhance the planning performance of the diffusion model. By utilizing the restoration gap metric and a gap predictor, the RGG process guides the reduction of the restoration gap and improves the quality of the generated plans.

Explainability: The paper highlights the explainability aspect of the approach. It presents the attribution maps of the gap predictor, which provide insights into error-prone transitions and enable a deeper understanding of the generated plans. Though, I believe this aspect needs to be evaluated further to prove its usefulness in different settings.

Empirical Evaluation: The proposed approach is evaluated on three different benchmarks in offline control settings, including Maze2D, Locomotion, and Block Stacking tasks. The experiments demonstrate the effectiveness of the approach in improving planning performance compared to other baseline methods.

Regularization Technique: To mitigate the risk of adversarial artifacts, the paper introduces an attribution map regularization method. This regularization prevents the gap predictor from pushing plans in the wrong direction, resulting in improved refining guidance.


**Weaknesses:**

This could be a weakness or a limitation (or a misunderstanding). If it’s limitation please elaborate on it in the Limitations section.
Basically, the proposed approach appears to assume that the diffusion model is well-trained and makes almost no mistakes. In the restoration gap part, the effectiveness of the "restoration gap" metric relies on the assumption that when a new trajectory is provided to the diffusion model to generate new plans, this trajectory data is assumed to have a similar distribution with the dataset which is used to train the diffusion model. In such cases, the restoration gap, based on L2 distance, might be applicable. However, if a new trajectory data which doesn't have a similar distribution is given to the diffusion model, the diffusion model might generate significantly different trajectories compared to the original trajectory. In this case, the restoration gap based on the L2 distance could potentially fail to provide accurate guidance. Can you comment on that?

Not weaknesses but just some small remarks are provided below:

I’d consider the third point in the contribution list as a result, and thus a verification of the proposed approach, not a contribution.

Line 226-227: I wouldn’t say RGG improves the planning performance of Diffuser in 5 out of 6 tasks confidently. 2-3 out 6 seems like a better judgment of these results.

Line 229: I think the claim “... our approach of refining Diffuser with RGG either matches or surpasses most of the offline RL baselines. Additionally, it significantly enhances the performance of Diffuser …” is too strong. Please be more precise and fair here. It would also help if you color-code the best- and second-best-achieving scores (surely while considering Diffuser, RGG and RGG+ as one algorithm, e.g., if RGG+ has the best score, do not consider RGG as the second-best if it has the highest score amongst others) in Table 1. That helps the reader put in perspective the performance of diffusion-based models to other RL approaches.

Line 235: repetition of a reference - “... and MOPO (Yu et al., 2020)”


**Questions:**

What can you say about the trade-off in terms of additional complexity and thus probably increased learning time, induced by the proposed extension, vs. the performance improvement?

I assume the lengths of the original trajectories and those generated by diffusion models are the same, right? If yes, are there additional steps to ensure this, or are they by default equal?

Maze environments might sometimes be solved using different trajectories. For instance, in Figure 2 (last row and last column maze), the generated trajectory takes the shortest path. However, it is possible for the diffusion model to take the longest path if the maze environment has been previously solved using longer trajectory paths. If these observations are stored in the dataset and used to train the diffusion model, it may generate longer trajectories but the original trajectory might be the shortest one. Is this a possible situation? If yes, how does this affect the restoration gap metric and would such a case cause it to inaccurately assess the quality of the generated paths?

What could be the reason(s) for the relatively smaller improvement on the Multi2D environment compared to a better improvement for the Maze2D scenario (Table 2)?

How is the performance score computed for Maze2D, Multi2D environments? E.g., the other score you shared for CQL seems to have been scored between 0 and 100 in the original paper (Fu et al., 2020). Your scores are above 100 though, how is this possible?


**Limitations:**

The authors briefly mentioned the limitations of the proposed approach in the appendix section. I believe the main text should include those limitation discussions.

In addition, if any of the questions / comments above relates to a limitation, please include them as well.

---

> ### Author Rebuttal · Authors · 2023-08-10
>
> **Q: What if the diffusion model is not well-trained? Can you comment on the accuracy of the restoration gap in such case?**
>
> **A:** We acknowledge that inaccuracies might arise especially in a low-data regime if the diffusion model is not good at estimating score functions for certain trajectories. However, when provided with sufficient data, the temporal compositionality property of Diffuser (Janner et al., 2022) stands out. This ensures that even with novel trajectories differing in distribution, the model can still generate accurate plans, leading to accurate estimation of restoration gap.
>
> This unique characteristic stems from the model’s ability to generate globally coherent trajectories by iteratively refining local consistency, allowing it to compose out-of-distribution trajectories from feasible subsequences. We will discuss this limitations especially in low-data regime, in our revised paper.
>
> Please also refer to our response to reviewer N3Wq regarding the potential inaccuracies of the restoration gap and our approach to mitigate them.
>
> **Q: The claim in Line 229 is too strong. It would be also helpful if you color-code the best- and second-best-achieving scores.**
>
> **A:** We agree that our claim is strong, we will tone down in the revised version. Moreover, we acknowledge the reviewer’s suggestion and we will make sure to incorporate the color-coding accordingly in Section 5.2.
>
> **Q: Trade-off in terms of additional complexity vs. the performance improvement?**
>
> **A:** As described in Appendix E.4, refining the diffusion planner requires the gap predictor for restoration gap estimation, using 500,000 synthetic data generated by Diffuser over 30 to 50 hours on a single NVIDIA RTX Quadro 8000 GPU, specifically in Block Stacking experiments. Subsequent gap predictor training on the same GPU takes 5 to 8 hours, which, in turn requires a maximum of 58 hours.
>
> However, we would like to highlight that this additional complexity remains tolerable, considering the benefits it yields. Notably, training the gap predictor allows to improve the performance across tasks, while also bringing an additional advantage in terms of injecting explainability into diffusion planners.
>
> **Q: I assume the lengths of the original trajectories and those generated by diffusion models are the same, right? If yes, are there additional steps to ensure this?**
>
> **A:** Yes, your assumption is correct. The lengths of the original trajectories and those restored trajectories should be the same to compute the restoration gap. However, as they are both generated from the same length of Gaussian random noise through the denoising process, there is no necessity for us to manipulate to ensure their equality in length.
>
> **Q: How do sub-optimal trajectories stored in the dataset affect the restoration gap metric?**
>
> A: It is essential to distinguish between two orthogonal aspects of the generated trajectories: **optimality** and **reliability**, wherein the restoration gap metric primarily pertains to reliability rather optimality.
>
> In general, optimality refers how efficient the trajectory is to solve a given task. For instance, in the context of Maze2D environments, the optimal trajectory refers the shortest path to reaches the goal position. As correctly pointed by the reviewer, the diffusion model may generate sub-optimal trajectories due to its learning from an offline dataset that contains sub-optimal trajectories. To enhance optimality, we can seek to maximize the cumulative reward along the trajectory. Notably, Diffuser employs return guidance during the denoising process to facilitate the maximization of cumulative rewards, represented as $\nabla\mathcal{J}$ in Equation 21.
>
> Reliability, on the other hand, pertains to the feasibility of the planned trajectory for the agent to execute. This reliability aspect is related to how likely the generated trajectory is aligned with the training data distribution, assuming a sufficient amount of dataset. To evaluate this feasibility, we first perturb the initially generated trajectory through the forward process. Subsequently, we restore the perturbed trajectory through the denoising process by the diffusion model. Given that infeasible trajectories violating physical constraints are absent from the training dataset, it is highly probable for the diffusion model to fail to restore the exact trajectory to the initially generated one. Thus, we facilitate the enhancement of reliability through restoration gap guidance during the denoising process, represented as $\nabla\mathcal{G}$ in Equation 21.
>
> **Q: What could be the reason(s) for the relatively smaller improvement on the Multi2D compared to the Maze2D?**
>
> **A:** Maze2D involves tasks that require navigating to a fixed goal located in a corner of the map. This setup, as compared to Multi2D, often leads to longer distances more detours around walls from the start to the goal, presenting more opportunities for generating infeasible plans, thereby accounting for the greater improvement observed in Maze2D.
>
> **Q: How is the score computed for Maze2D and Multi2D?**
>
> **A:** A reward of 0 is given for each timestep, and a reward of 1 is received once the goal is reached. Therefore, the quicker the goal is reached, the higher the sum of rewards will be. Furthermore, in accordance with Janner et al. (2022), we normalize the score as follows:
>
> $$
> \text{normalized score} = \frac{(\text{score} - 6.7)}{(273.99 - 6.7)}*100
> $$
>
> **Q: Main text should include limitation discussions.**
>
> **A:** We inevitably describe limitation of our work in Appendix section due to space constraint. We will include limitation in main text in camera-ready version.
>
> **References**
>
> - Janner et al., Planning with Diffusion for Flexible Behavior Synthesis. In *ICML*, 2022.

---

> > ### Comment · Reviewer_mq5w · 2023-08-14
> >
> > I appreciate the clarifications by the authors, I will surely consider these points in my final evaluation.

---

> > > ### Author Response · Authors · 2023-08-20
> > >
> > > Thank you for acknowledging the novelty and contributions including performance improvement and explainability of our work. In the final version, we’ll incorporate the clarifications we mentioned in our rebuttal into the paper. Furthermore, we will tone down our claim regarding the planning performance and integrate the content from the Limitations section in the Appendix into the main body of the paper.
> > >
> > > If you have any additional suggestions or concerns, please feel free to share them with us before the discussion period ends.
> > >
> > > Once again, we extend our gratitude for your invaluable suggestions and comments.

---

### Official Review · Reviewer_pN8B · 2023-07-07

**Soundness:** 3 good
**Presentation:** 3 good
**Contribution:** 3 good
**Rating:** 8
**Confidence:** 3

**Summary:**

The paper proposes a novel metric named restoration gap for evaluating trajectory quality generated by diffusion models. It further proposed restoration gap guidance to refine the diffusion generation. The restoration gap predictor also provides explainability of the model. Experiment results indicate improved performance of the proposed model.

**Strengths:**

The contribution of the paper is solid: The restoration gap gives an intuitive metric for measuring trajectory quality: plans with low quality are hard to restore after perturbation; Following this intuition, the author proposed to train a gap predictor and uses its gradient to guide the generation process. Experiment results give strong support for the claim of the paper.

The presentation of the paper is clear. The paper is well-structured with intuitive visualizations.



**Weaknesses:**

To apply the restoration gap guidance, a hyperparameter is introduced to weight the signal from the gap predictor; and then to mitigate the issue from the gap prediction error, an attribution map regularization is added with another hyperparameter weight. It is not clear how sensitive the performance of the model is to these hyperparameters and how hard it is to tune them.


**Questions:**

Can author give some results / explainations on how sensitive the performance of the model is to the additional hyperparameters introduced (beta, lambda in eq. 22) and how hard it is to tune them?

**Limitations:**

While the restoration gap is intuitive in the given examples (2D map with walls), it is unclear how the gap works in more complex environments or environments with fewer hard constraints.

---

> ### Author Rebuttal · Authors · 2023-08-09
>
> **Q: While the restoration gap is intuitive in the given examples (2D map with walls), it is unclear how the gap works in more complex environments or environments with fewer hard constraints.**
>
> **A:** In more complex environments with fewer hard constraints, determining infeasibility might not be as straightforward as detecting a plan that involves going through a wall in the Maze2D environment. However, our approach remains valid as the concept of infeasibility can be extended to less constrained environments.
>
> In the HalfCheetah environment, for instance, a plan could be seen as infeasible if it entails abrupt changes in the direction or speed of the cheetah, which would be physically improbable. Similarly, in the Unconditional Block Stacking environment, plans that involve the robotic arm suddenly teleporting or grasping a block stacked under other blocks could be considered infeasible.
>
> We reported detailed visualizations and explanations of such instances in Appendix C.4. The plans with high restoration gaps often exhibit such physically improbable transitions in these environments, and our restoration gap measure correctly identifies these as unreliable.
>
> **Q: Can author give some results / explanations on how sensitive the performance of the model is to the additional hyperparameters introduced (beta, lambda in eq. 22) and how hard it is to tune them?**
>
> **A:** Please refer to the global response provided above.

---

### Official Review · Reviewer_N3Wq · 2023-07-09

**Soundness:** 3 good
**Presentation:** 3 good
**Contribution:** 3 good
**Rating:** 7
**Confidence:** 3

**Summary:**

This paper introduces a novel approach to refining unreliable plans generated by diffusion models in long-horizon, sparse-reward tasks. While diffusion-based planning has shown promise, the generative nature of diffusion models can result in infeasible plans, limiting their usefulness in safety-critical applications. The proposed approach utilizes a metric called restoration gap to evaluate plan quality and provides refining guidance to error-prone plans. Additionally, an attribution map regularizer is presented to prevent adversarial refining guidance. The effectiveness of the approach is demonstrated on three benchmarks, and the explainability of the generated plans is highlighted through the attribution maps.

**Strengths:**

1. Detailed and comprehensive experiments: The article provides a thorough evaluation of the proposed approach on three different benchmarks in offline control settings that require long-horizon planning. This comprehensive experimentation strengthens the validity of the findings.
2. Novel idea and solution: The article introduces a new metric, restoration gap, and a corresponding refining guidance approach to address the limitations of diffusion-based planning. This novel approach contributes to the field and expands the understanding of plan refinement in such tasks.
3. Addressing the issue of error-prone plans: The article successfully tackles the challenge of refining unreliable plans generated by diffusion models. By providing refining guidance based on the restoration gap metric, the proposed approach improves the feasibility and quality of the plans.

**Weaknesses:**

1） Accuracy of restoration gap metric: This paper uses the difference between the generated trajectory and the original input trajectory to evaluate plan quality. However, this approach may be inaccurate for small-sample cases where the distribution of the expert dataset leads to rare but valid trajectories being considered abnormal. Further exploration is needed to address this limitation.
2） Lack of experimentation on balancing coefficients: The article does not conduct an ablation study to examine the balance between the opportunity cost function J's guide and the opportunity restore guide. It would be beneficial to investigate how to strike a balance between these coefficients to achieve optimal plan refinement.

**Questions:**

1）How does the proposed approach address the potential issue of rare but valid trajectories being considered abnormal due to the distribution of the expert dataset? Have any measures been taken to mitigate this limitation?
2）Could you provide more insight into how the coefficients for the opportunity cost function J's guide and the opportunity restore guide are balanced in the proposed approach? Has any experimentation been conducted to determine the optimal balance between these coefficients?

**Limitations:**

the article should address the issues related to the accuracy of the restoration gap metric in few-shot case and conduct experiments to explore the balance between coefficients in the opportunity cost function and the opportunity restore guide.

---

> ### Author Rebuttal · Authors · 2023-08-09
>
> **Q: How does the proposed approach address the potential issue of rare but valid trajectories being considered abnormal due to the distribution of the expert dataset?**
>
> **A:** We appreciate the reviewer’s insightful comment regarding the accuracy of the restoration gap. Indeed, the inaccuracy comes from the two cases: 1) when the initially generated trajectory (the original input trajectory) is valid but considered invalid, and 2) when the initially generated trajectory is invalid but incorrectly considered valid. We acknowledge that the reviewer’s specific concern pertains to the first case, where valid trajectories may be misclassified.
>
> To address this issue, we employ a strategy that involves sufficiently corrupting the given trajectory (i.e., the originally generated trajectory) during the restoration process by setting a large enough value for $\hat{t}$, where $\hat{t} \in (0, 1]$ is a continuous time variable for indexing diffusion timestep. As the expert dataset does not contain invalid trajectories at all, valid trajectories inherently possess larger density compared to invalid ones, even though their density might be relatively small. Therefore, by perturbing the given trajectory significantly, we ensure that the restored trajectory from an invalid trajectory diverges more significantly in terms of the L2 distance compared to the restored trajectory from a valid trajectory, through the extended denoising procedure.
>
> We have conducted a theoretical analysis, outlined in Proposition 1, which supports this approach. The first case error that the reviewer highlights corresponds to the type I error (false positive). According to Proposition 1, the error probability of restoration gap for type I error is bounded at most $\delta$ through thresholding with a threshold value of $b_I = \sigma_{\hat{t}}\left(C\sigma_{\hat{t}} + \sqrt{d} + \sqrt{d + 2 \sqrt{-d \cdot \log \delta} - 2 \log \delta)}\right)$, under specific regularity conditions. To tightly bound the type I error, it is crucial to have a large threshold value $b_I$, requiring a sufficiently large $\sigma_{\hat{t}}$, which, in turn, implies employing a large enough $\hat{t}$. For further details, please refer to Appendix B.
>
> Please also note that we conducted an ablation study on the sensitivity to perturbation magnitude $\hat{t}$ in Appendix C.2 which stands for this theoretical analysis. Figure 6 in Appendix C.2 demonstrates that the performance of plans from the restoration gap when setting $\hat{t}=0.9$ performs the best among $\hat{t}=0.3, 0.5, 0.7, \text{and } 0.9$.
>
> We thank the reviewer for bringing up this issue, and we will explicitly state this aspect in the revised version.
>
> **Q: Lack of experimentation on balancing coefficients.**
>
> **A:** Please refer to the global response provided above.

---

> > ### Comment · Reviewer_N3Wq · 2023-08-18
> > **Thanks for the author's patient response**
> >
> > Thanks for the author's patient response. I believe the proposed solution and theoretical analysis are both effective and correct.

---

> > > ### Author Response · Authors · 2023-08-20
> > >
> > > We are glad to hear that our rebuttal was helpful and that you find both the proposed solution and theoretical analysis effective and correct. In response to your feedback, we will provide insights into the accuracy of the restoration gap through theoretical analysis in the revised paper. Furthermore, we will elaborate on the sensitivity of the planning performance in relation to the choice of scaling coefficients, along with an explanation of our tuning process.
> > >
> > > If you have any remaining suggestions or concerns, please let us know before the discussion period ends.
> > >
> > > Thank you once again for your invaluable suggestions and comments.

---

### Official Review · Reviewer_iw4y · 2023-07-28

**Soundness:** 3 good
**Presentation:** 2 fair
**Contribution:** 3 good
**Rating:** 7
**Confidence:** 4

**Summary:**

The paper proposes to refine the predictions of a diffusion-based planner by identifying unfeasible plans prior to execution. The newly introduced "restoration gap" measures the reconstruction error for a given sample under some noisy perturbation, and it is argued that this gap is a reasonable proxy for feasibility (i.e. if the restoration gap is high for a sample (plan), then that plan is probably not feasible). The authors show that the proposed metric can identify "artifacts" with bounded error probabilities. The proposed approach trains a separate model known as the gap predictor, which simply predicts the restoration gap for a given sample. Then this gap predictor, which is differentiable, can be used to guide the diffusion sampling process away from unfeasible plans. Additionally, since errors in the gap predictor could lead to model exploitation, the authors propose to use attribution maps to perform regularization. Experiments on various offline control tasks (Maze2D, Mujoco locomotion, Block stacking) show that the proposed approaches (with/without regularization) outperform several strong offline RL baselines. The authors also show how the attribution maps can be used to identify specific parts of samples which contribute the most to the restoration gap, which is useful for identifying problematic transitions for instance.

**Strengths:**

The proposed work seems novel and the experimental results show that the proposed work is state-of-the-art.

From my readthrough, I didn't identify any glaring technical flaws and the approach itself seems sound.

Section 5.3 is very solid and highlights a pretty interesting application of the proposed approach that I haven't seen before.

**Weaknesses:**

I disagree a bit on the high-level motivation for this work. I don't think there is anything about the restoration gap that specifically targets "unfeasible" plans (by "unfeasible" I mean plans that we cannot execute due to certain constraints, which is different from plans that are just bad under some reward function). For instance, I would expect that highly OOD plans have very high restoration gaps even if they are technically feasible to execute, because our diffusion model will struggle to reconstruct plans that are not in the training distribution. My suspicion is that this approach is equivalent to performing some kind of low temperature sampling to ensure predicted samples are high confidence. If that is the case, then I think the paper needs some rewriting because the approach really is not related to feasibility I think. Would be curious to hear if the authors have a counterpoint to this, but otherwise I think this should be addressed.

Since I do think this is fairly major concern with the writing, my score is a bit lower, but I still think the core technical contribution of this paper is reasonable so I'm happy to bump my score if the paper is revised, or it turns out that I am incorrect in my suspicions.

The experimental results aren't terribly impressive, in most settings the proposed approach is barely better than Diffuser.

**Questions:**

Is it possible to adapt the metrics compared to in 5.1 to downstream control? I think it would be helpful if the authors could show that the proposed restoration gap metric is actually better than prior metrics on relevant control benchmarks. I'm not super familiar with these other metrics so I don't know how feasible that is.

I would suggest including more formal definitions for "faithful" and "infeasible" plans. I have no idea what a faithful plan is. I think I know what an infeasible plan is, but given the aforementioned confusing around the motivation for the paper, it's possible I have a different definition in mind.

Also for some reason I am unable to open the supplementary materials zip file, so nothing in there affected my review.

===
I have read the rebuttal and bumped my score up. I am in favor of acceptance.

**Limitations:**

Yes.

---

> ### Author Rebuttal · Authors · 2023-08-09
>
> **Q: I disagree a bit on the high-level motivation for this work. I don’t think there is anything about the restoration gap that specifically targets “unfeasible” plans.**
>
> **A:** In low-data regime, we acknowledge the scenario where even feasible plans can appear highly OOD, leading to a substantial restoration gap. However, our high-level motivation mainly targets scenarios where there is a sufficient offline dataset comprising feasible trajectories. The property of **temporal compositionality** in Diffusion Planners, as described by Janner et al. (2022), plays a significant role in this context.
>
> With temporal compositionality, Diffusion Planners can generate globally coherent trajectories by continuously enhancing local consistency. This allows them to compose out-of-distribution trajectories by stitching together any feasible plan subsequences.
>
> Given this property, we argue that most infeasible plans, those that cannot be executed due to system constraints, would likely be OOD, while feasible plans would generally remain within the distribution. So, the restoration gap, in our view, does hold relevance in identifying unfeasible plans. We appreciate your thoughtful feedback and we will make sure to emphasize this context in our revision.
>
> **Q: I would suggest including more formal definitions for “faithful” and “infeasible” plans.**
>
> **A:** In our context, “faithful” refers to plans that are consistent with known constraints and can be executed. Conversely, “infeasible” plans are those that cannot be executed due to system constraints. We will include formal definitions on the revised version for clarity.
>
> **Q: Also for some reason I am unable to open the supplementary materials zip file, so nothing in there affected my review.**
>
> **A:** We sincerely apologize for the inconvenience you experienced in accessing the supplementary materials. We have tested the accessibility of the zip file across various OS, including Windows, Ubuntu, and MacOS where it worked fine for all systems. It appears that the issue might be specific to your system configuration. If you provide additional details regarding the problem you encountered, we’ll address the problem.
>
> In the meantime, with the approval of the Area Chair, we have provided an anonymized Dropbox link containing the exact files uploaded on OpenReview. You can access them directly using the following link: [Supplementary Materials](https://www.dropbox.com/scl/fo/2dwyps3kw02eveyyog9ue/h?rlkey=dv07r9vp5eccj9s172egn76j6&dl=0). In our final submission, we’ll ensure the materials are accessible.
>
> **Q: The experimental results aren’t terribly impressive.**
>
> **A:** Please refer to the global response provided above.
>
> **Q: Is it possible to adapt the metrics compared to in 5.1 to downstream control?**
>
> **A:** Please refer to the global response provided above.
>
> **References**
>
> - Janner et al., Planning with Diffusion for Flexible Behavior Synthesis. In *ICML*, 2022.

---

> > ### Comment · Reviewer_iw4y · 2023-08-12
> >
> > I appreciate the comprehensive rebuttal from the authors.
> >
> > On the topic of experimental results, I think the empirical results are more impressive with the added context.
> >
> > As for the high-level motivation, I think we're on the same page about the fact that the approach targets unfeasible trajectories using "OOD"-ness, where the paper argues (and empirically supports) that when trained on expert datasets, unfeasible trajectories are likely to be OOD for our model. I would still suggest that the authors be careful when doing revisions to distinguish being feasible and being OOD since they are definitely different, although I agree that this is a useful heuristic and would be of interest to the research community.
> >
> > Also, thanks for the Dropbox link. Not sure what the issue was, but much appreciated.
> >
> > With all that being said, I'll bump my score up and recommend acceptance.

---

> > > ### Author Response · Authors · 2023-08-20
> > >
> > > We are pleased to hear that our rebuttal effectively addressed your concerns, and we are sincerely grateful for the increased score and your recommendation for acceptance. We will make sure to clarify what unfeasible trajectories are and discuss how they relate to OOD plans in the final version of our paper.
> > >
> > > If you have any remaining suggestions or concerns, please let us know before the discussion period ends.
> > >
> > > Once again, we extend our gratitude for your invaluable suggestions and comments.

---

### Author Rebuttal · Authors · 2023-08-09

We would like to sincerely thank all reviewers for their time and effort in reviewing our paper, and for their constructive feedback and comments. In this thread, we answer commonly asked questions. Please find below our response to your insightful comments and suggestions.

**Q: Lack of substantial improvement demonstrated by the proposed solution compared to the Diffuser benchmark. (to Reviewer iw4y and 9DAN)**

**A:** We would like to clarify that the planning performance improvement by restoration gap guidance (RGG) is not trivial. To the best of our knowledge, concurrent works to our approach that followed Diffuser (Janner et al., 2022) can be summarized into two methods: 1) Decision Diffuser (Ajay et al., 2023) and 2) AdaptDiffuser (Liang et al., 2023), the latter of which we have recently discovered at ICML 2023.

When comparing RGG to Decision Diffuser, we find that RGG performs on par with a slight difference. For instance, in Locomotion tasks (Fu et al., 2020), the average scores achieved by RGG and RGG+ are 81.2 and 81.6, respectively, while Decision Diffuser reaches a score of 81.8. However, Decision Diffuser necessitates training the diffusion model from scratch through classifier-free guidance with low temperature sampling, while our RGG does not require such a process and can be applied to any unconditional diffusion-based planner which offers a chance to further refinement of generated sequence. Moreover, explicitly modeling the restoration gap predictor in our approach enables the diffusion planner to additionally have explainability, facilitating the identification of error-prone transitions within infeasible trajectories. These distinctions make the performance achievement of RGG non-trivial.

Regarding the comparison with AdaptDiffuser, we acknowledge that the planning performance of RGG and RGG+ is slightly lower but still comparable. It is crucial to note that AdaptDiffuser assumes knowledge of the oracle inverse dynamics for collecting additional training data, which is a quite strong assumption and uncommon in general machine learning scenarios. Considering this, we firmly assert that our achievement in performance is indeed significant, as it does not rely on such specific and uncommon assumptions.

**Q: Is it possible to adapt the metrics compared to in 5.1 to downstream control? (to Reviewer iw4y and 9DAN)**

**A:** To directly compare the performance of plans guided by different metrics, we run new experiments in the Maze2D Large and Multi2D Large environments. The results are as follows:

| Environment | Diffuser | Rarity | Negative Realism | RGG | RGG+ |
|:--------------:|:--------:|:------:|:----------------:|:-----:|:-----:|
| Maze2D Large | 123.5 | 126.9 | 128.9 | **135.4** | **143.9** |
| Multi2D Large | 141.2 | 143.4 | 143.3 | **148.3** | **150.9** |

The result clearly shows that the restoration gap is a useful metric for control tasks. Unlike other metrics that need expert data for training, our restoration gap works without such constraints, making it even more practical.

**Q: Can author give some results / explanations on how sensitive the performance of the model is to the additional hyperparameters introduced (beta, lambda in eq. 22) and how hard it is to tune them? (to Reviewer N3Wq and pN8B)**

**A:** The planning performance of RGG and RGG+ is somewhat sensitive to these hyperparameters. In Figure 3, we can observe implicit indications of sensitivity, where the performance varies depending on the choice of the value of $\lambda$.

However, we would like to highlight that there exists an evident tendency in the change of planning performance relative to the selected values of these hyperparameters, making it manageable to tune them effectively through a grid search. For example, Figure 3 illustrates that the planning performance increases as the value of $\lambda$ rises, but then declines for excessively large $\lambda$ values.

In our experiments, we conducted a grid search for the hyperparameters where the search spaces are presented in Table 10, 11, 12, 13, and 14, for Maze2D, HalfCheetah, Hopper, Walker2D, and Block Stacking experiments, respectively. The planning performance exhibited a clear trend based on the choices of $\alpha$, $\beta$, and $\lambda$, thus requiring only a small number of evaluation episodes for the grid search. For instance, we used 10, 15, and 10 evaluation episodes for Maze2D, Locomotion, and Block Stacking experiments, respectively.

We genuinely appreciate the reviewer’s valuable feedback, and we agree that including an ablation study on the sensitivity of the planning performance to the scaling coefficients would enhance the thoroughness of our work. We will incorporate this study in the revised version to provide further clarify on this aspect.

**References**

- Janner et al., Planning with Diffusion for Flexible Behavior Synthesis. In *ICML*, 2022.
- Ajay et al., Is Conditional Generative Modeling All You Need for Decision-Making? In *ICLR*, 2023.
- Liang et al., AdaptDiffuser: Diffusion Models as Adaptive Self-evolving Planners. In *ICML*, 2023.
- Fu et al., D4rl: Datasets for Deep Data-Driven Reinforcement Learning. *arXiv*, 2020.

---

### Decision · Program_Chairs · 2023-09-21

**Decision:**

Accept (poster)

**Comment:**

Authors propose an interesting restoration gap metric and predictor to help refine diffusion planners towards more feasible path generations (not going through obstacles etc). While some reviewers initially had some concerns, the authors have taken care to address all concerns, and reviewers have larger accepted all these rebuttals. This paper is a clear accept.